# Efficient electrosynthesis of formamide from carbon monoxide and nitrite on a Ru-dispersed Cu nanocluster catalyst

Jiao Lan[1,4], Zengxi Wei[2,4], Ying-Rui Lu [3,4], DeChao Chen[1], Shuangliang Zhao [2], Ting-Shan Chan [3] ✉ & Yongwen Tan [1] ✉

Conversion into high-value-added organic nitrogen compounds through electrochemical C-N coupling reactions under ambient conditions is regarded as a sustainable development strategy to achieve carbon neutrality and high-value utilization of harmful substances. Herein, we report an electrochemical process for selective synthesis of high-valued formamide from carbon monoxide and nitrite with a $Ru_1Cu$ single-atom alloy under ambient conditions, which achieves a high formamide selectivity with Faradaic efficiency of $45.65 \pm 0.76\%$ at $-0.5$ V vs. RHE. In situ X-ray absorption spectroscopy, coupled with in situ Raman spectroscopy and density functional theory calculations results reveal that the adjacent Ru-Cu dual active sites can spontaneously couple *CO and *$NH_2$ intermediates to realize a critical C-N coupling reaction, enabling high-performance electrosynthesis of formamide. This work offers insight into the high-value formamide electrocatalysis through coupling CO and $NO_2^-$ under ambient conditions, paving the way for the synthesis of more-sustainable and high-value chemical products.

Amides and their derivatives represent commercially important category of organic compounds as witnessed by their widespread use as intermediates in the manufacture of chemical as well as polymer and biological compounds[1–7]. Among them, as an important chemical raw material, formamide ($HCONH_2$) has been widely used in organic synthesis, pharmaceuticals, plastics, and pesticides, etc.[5–7]. At present, however, the industrial synthesis of formamide is generated by the reaction of fossil fuel-derived carbon monoxide (CO) and ammonia ($NH_3$) under harsh reaction conditions, resulting in a large amount of energy consumption and emission of a large amount of greenhouse gases[6,7]. Moreover, special equipment and complicated synthetic processes are often required to improve the limited conversion efficiency[2,4,6]. Therefore, it is desirable to develop sustainable routes that enable efficient and low-cost synthesis of formamide under milder conditions.

Conversion of high value-added organonitrogen compounds by electrochemical C–N coupling reaction under environmental conditions is considered as a sustainable strategy to achieve carbon neutrality and high-value utilization of hazardous substances[8–15]. Recent advances demonstrate that electrochemical synthesis of organic amides from C–N coupling reaction by introducing an $NH_3$ source during $CO_2$/CO reduction reaction[14,16]. For example, Jiao et al. reported that nucleophilic addition of $NH_3$ could boost the reduction of CO to *C=C=O, which in turn reacts with $NH_3$ to form intermediates that proceed to form acetamide under strong basic conditions, demonstrating the potential for generating amide formation[14]. However, more advanced catalytic processes are still needed to expand the scope of possibilities of catalyzed C–N bond formation for generating more valuable products.

In this regard, nitrite/nitrate ($NO_2^-$/$NO_3^-$) is a highly abundant nitrogen source, particularly in industrial wastewater and polluted

[1]College of Materials Science and Engineering, State Key Laboratory of Advanced Design and Manufacturing for Vehicle Body, Hunan University, Changsha, Hunan 410082, China. [2]Guangxi Key Laboratory of Petrochemical Resource Processing and Process Intensification Technology and School of Chemistry and Chemical Engineering, Guangxi University, Nanning 530004, China. [3]National Synchrotron Radiation Research Center, Hsinchu 300, Taiwan. [4]These authors contributed equally: Jiao Lan, Zengxi Wei, Ying-Rui Lu. ✉e-mail: chan.ts@nsrrc.org.tw; tanyw@hnu.edu.cn

groundwater[17,18]. Moreover, the electrochemical $NO_2^-$ reduction reaction ($NO_2^-$RR) offers a practical path to product $NH_3$ with renewable electricity due to the lower dissociation energy for the N=O bond (204 kJ mol⁻¹)[19,20]. Inspiring by this, electrocatalytic coupling $NO_2^-$ with CO might be an alternative route to drive formamide synthesis by using abundant and cheap C- and N-containing feedstocks. The key challenges are the rational design of efficient and stable active sites for C/N precursor reduction and C–N coupling to improve formamide selectivity.

Herein, we realize the electrochemical coupling of CO with $NO_2^-$ to product formamide with an electrocatalyst consisting of atomically dispersed Ru atoms on Cu nanoclusters single-atom alloy (denoted $Ru_1Cu$ SAA). Direct experimental evidence shows that isolated Ru atoms is incorporated into the lattice of Cu nanoclusters to generate $Ru_1Cu$ SAA. The $Ru_1Cu$ SAA affords an average formamide yield rate of $2483.77 \pm 155.34$ μg h⁻¹ mg$_{cat.}$⁻¹, as well as ultra-high Faradaic efficiency (FE) of $45.65 \pm 0.76\%$ at −0.5 V vs. reversible hydrogen electrode (RHE). Furthermore, a series of in situ experimental studies and theoretical calculations unveil that the adjacent Ru–Cu dual active sites act as intrinsic active centers: the single Ru atom promotes the adsorption of $NO_2^-$ and activates the deoxygenation hydrogenation process, while

CO undergoes dissociation adsorption on adjacent Cu atoms. Consequently, the Ru–Cu synergistic catalysis promotes the formation of C–N bond through spontaneous coupling of *CO and *NH₂ intermediates, resulting in high activity and selectivity toward electro-synthesis of formamide. Compared with monometallic catalysts, the dual-active-site catalyst can provide independent dual-site synergistic catalysis for C and N activation, thereby enhancing the C–N coupling efficiency.

## Results

### Structural characterization

Figure 1a shows the schematic diagram of the fabrication process of $Ru_1Cu$ SAA (see "Methods"). First, the $TiO_2$ nanowires with a diameter of ~10 nm were easily fabricated by chemical dealloying method served as catalyst supports (Supplementary Fig. 1 and Supplementary Fig. 2a)[21]. Subsequently, Cu nanoclusters (denoted as Cu NCs) were loaded onto the dealloyed $TiO_2$ nanowires by thermal reduction. Afterward, $Ru_1Cu$ SAA was obtained by introducing Ru single atom onto the surface of Cu NCs through galvanic replacement reaction (Fig. 1b)[22]. Meanwhile, RuCu alloy (denoted as RuCu NPs) samples with higher Ru doping levels were synthetized as control samples. It is

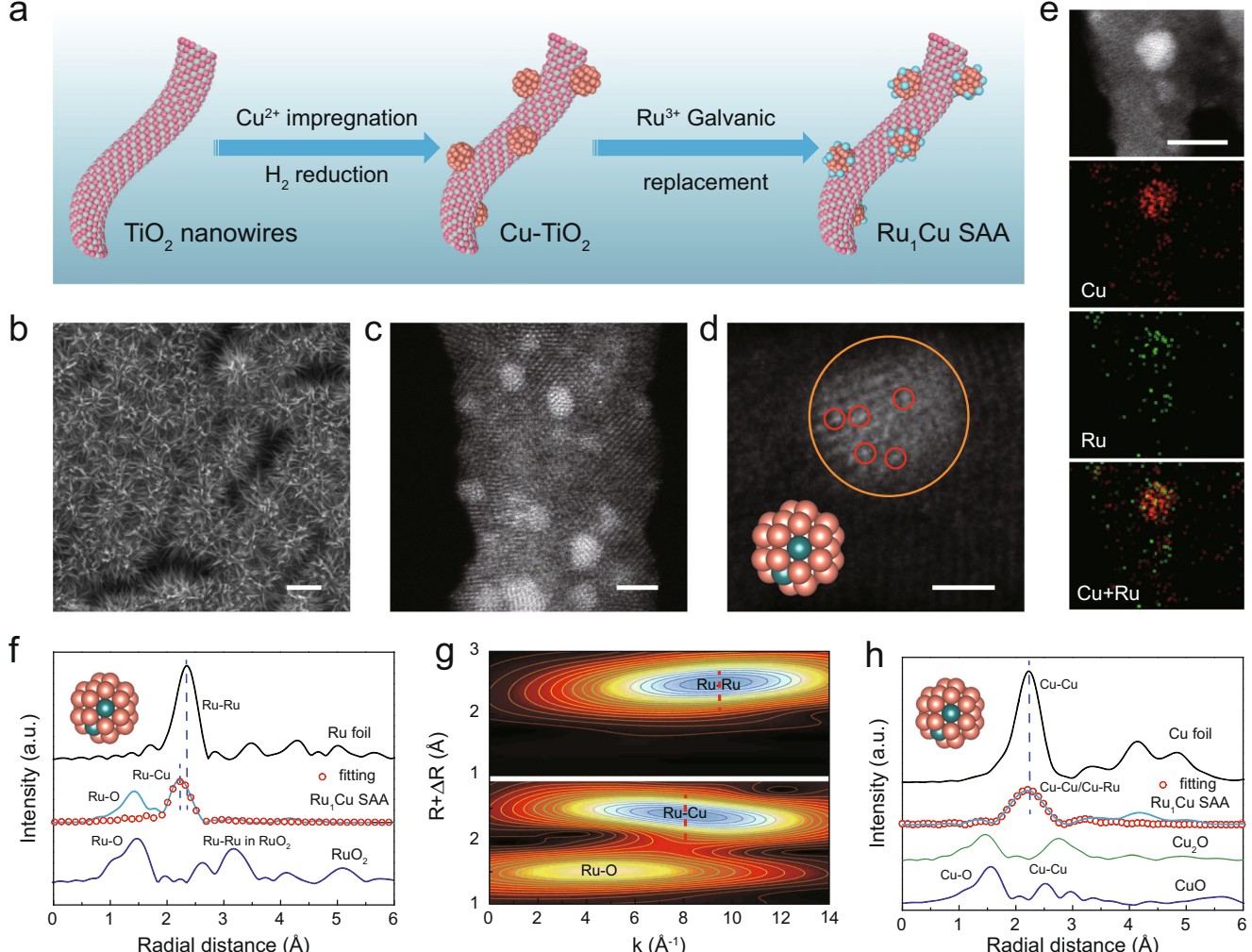

**Fig. 1 | Structural characterizations of Ru₁Cu SAA. a** Schematic illustration of preparation processes for Ru₁Cu SAA catalysts, with Ti, O, Ru, and Cu atoms shown as pink, gray, blue, and orange, respectively. **b** SEM image of the Ru₁Cu SAA. **c** HAADF-STEM image of Ru₁Cu SAA. **d** Ru₁Cu SAA enlarged images. **e** HAADF-STEM image and the corresponding elemental mapping. **f** Ru K-edge FT-EXAFS spectra of Ru₁Cu SAA and reference samples (RuO₂, and Ru foil), and corresponding Ru₁Cu SAA fitting curves, inset showing the schematic model. **g** Ru K-edge EXAFS WT analysis of Ru₁Cu SAA and Ru foil. **h** Cu K-edge FT-EXAFS spectra of Ru₁Cu SAA and reference samples (Cu₂O, CuO, and Cu foil), and corresponding Ru₁Cu SAA fitting curves, inset showing the schematic model. Scale bars: **b** 200 nm, **c** 2 nm, **d** 1 nm, **e** 5 nm.

noted that Cu NCs and $Ru_1Cu$ SAA show similar X-ray diffraction (XRD) patterns without Ru or $RuO_2$ phase, implying a high dispersion degree of Ru species (Supplementary Fig. 2a). In contrast, the lattice constants of RuCu NPs increase with the increase of Ru loading (Supplementary Fig. 2b) due to the substitution of Cu by Ru with a larger atomic radius. Low-magnification scanning transmission electron microscopy (STEM) image shows that homogeneous $Ru_1Cu$ SAA with an average size (-1.5 nm) are well dispersed and anchored onto the dealloyed $TiO_2$ nanowires support (Supplementary Fig. 3). The high-angle annular dark-field scanning transmission electron microscopy (HAADF-STEM) (Fig. 1c) image clearly displays that number of bright and atom-sized features attributed to individual Ru atoms can be discerned on the crystal surface of Cu in $Ru_1Cu$ SAA. It is noteworthy that these isolated Ru atoms are surrounded by Cu atoms in different regions of the $TiO_2$ nanowires without agglomerating into Ru nanoclusters. More importantly, the randomly magnified image further confirms the substitution of surface Cu atoms by isolated Ru atom (Fig. 1d), where the single Ru atoms (red circle) are located on Cu NCs, demonstrating the formation of $Ru_1Cu$ single-atom alloy. The STEM-coupled energy dispersive spectroscopic (EDS) elemental mapping reveals a uniform dispersion of Ru dopants in the Cu nanocluster matrix in $Ru_1Cu$ SAA (Fig. 1e), with a Ru:Cu atomic ratio of approximately 4:96 (Supplementary Fig. 4), which is consistent with inductively coupled plasm optical emission spectroscopy (ICP-OES) results (Supplementary Table 1). Furthermore, low-magnification STEM images show that the average size of RuCu NPs is -1.1 nm, and the energy dispersive spectroscopy (EDS) confirmed that Cu and Ru are uniformly distributed in RuCu NPs (Supplementary Fig. 5).

The electronic structure and surface composition of Cu NCs, $Ru_1Cu$ SAA, and RuCu NPs were investigated by X-ray photoelectron spectroscopy (XPS). The two paired peaks of Ru 3$d$ XPS spectra are deconvoluted for $Ru^{0+}$ and $Ru^{4+}$ species (Supplementary Fig. 6a). Notably, the binding energy of $Ru_1Cu$ SAA shifts toward high binding energy by -0.54 eV as compared with that of RuCu NPs, indicating that the Ru species carry more positive charges to assume the oxidation state[23]. Similarly, in the Cu 2$p$ region, two paired peaks are associated with $Cu^{0/1+}$ and $Cu^{2+}$ species (Supplementary Fig. 6b). Moreover, the binding energy of Cu 2$p_{3/2}$ in the $Ru_1Cu$ SAA (932.16 eV) shifts negatively (-0.37 eV) compared with the Cu NCs (932.53 eV). The shift of the binding energy is ascribed to the electronic interaction and charge transfer between Ru and Cu[13,23]. Furthermore, we used X-ray absorption spectroscopy (XAS) to further confirm the atomically dispersed Ru and probe the electronic and coordination structure of $Ru_1Cu$ SAA. The Ru K-edge extended X-ray absorption near-edge structure (XANES) spectrum of $Ru_1Cu$ SAA exhibits a distinct energy absorption edge profile compared with Ru foil (Supplementary Fig. 7), indicating the formation of Ru oxidized state in Ru $_1$Cu SAA due to slight oxidation of Ru atoms[24,25]. The corresponding Fourier-transformed extended X-ray absorption fine structure (FT-EXAFS) spectrum of the $Ru_1Cu$ SAA shows two distinct peaks (1.42 Å and 2.22 Å) (Fig. 1f). The obvious 1.42 Å peak ascribed to Ru−O scattering contributions, which is mainly caused by the inevitable oxidation of the material in the air and the loading of some Ru atoms on the $TiO_2$ substrate during the material synthesis process[24,26]. While the peak at 2.22 Å in $Ru_1Cu$ SAA is distinct from that in Ru foil (2.39 Å), which could be tentatively assigned to Ru−Cu contribution[27], indicating that the Ru dopants were atomically dispersed in Cu matrix after the galvanic replacement process and did not form nanocluster[28]. The presence of Ru−Cu scattering is further corroborated by wavelet transforms (WT) of Ru EXAFS oscillation in Fig. 1g. Besides the Ru−O bond (4.6 Å$^{-1}$), one intensity maximum at near 8.1 Å$^{-1}$ is exclusively observed, which is assigned to the Ru−Cu contribution in $Ru_1Cu$ SAA, clearly different from that in Ru foil, suggesting the formation of atomically dispersed Ru atoms on the Cu nanocluster[22]. Then according to a primitive model optimized with density functional theory (DFT, Supplementary Fig. 8), $Ru_1Cu$ SAA was constructed by replacing two Cu atoms with two Ru atoms on the $Cu_{38}$ cluster model, where two Ru atoms are located at the centers of two adjacent Cu hexagons. A least-squares EXAFS fitting analysis for R-space spectrum of Ru in $Ru_1Cu$ SAA (Fig. 1f and Supplementary Table 2) indicates that the coordination number (CN) of the center Ru atom with surrounding Cu atoms on $Ru_1Cu$ SAA is approximately 4.8. This further confirms the presence of Ru−Cu coordination in $Ru_1Cu$ SAA, consistent with the WT analysis results[29]. The results suggest that predominant Ru are distributed as isolated and did not form nanoclusters. Additionally, Cu K-edge XANES spectra and the corresponding FT-EXAFS of $Ru_1Cu$ SAA show very similar absorption edge and peak positions with that of Cu foil (Supplementary Fig. 9 and Fig. 1h). However, the lower Cu−Cu/Cu−Ru scattering intensity of the $Ru_1Cu$ SAA compared to the Cu foil indicates the obvious local unsaturated coordination of the $Ru_1Cu$ SAA (Fig. 1h and Supplementary Table 2), which possesses higher catalytic activity[30]. Therefore, both HAADF-STEM, XAS characterizations, and structural modeling studies demonstrate the formation of $Ru_1Cu$ SAA with atomically dispersed Ru atoms on the surface of Cu nanoclusters. Such $Ru_1Cu$ SAA with dual sites is expected to provide flexible adsorption configurations for reaction intermediates and facilitate the coupling of C/N intermediates[31,32].

## Electrochemical performance

The electrocatalytic performance evaluation of $Ru_1Cu$ SAA was carried out under ambient conditions using the chronoamperometry method in a standard three-electrode electrochemical device. Gaseous products were quantified by gas chromatography (GC) (Supplementary Fig. 10) and liquid products were quantified using nuclear magnetic resonance (NMR) (Supplementary Figs. 11 and 12) and colorimetric analysis (Supplementary Fig. 13). The intrinsic CO reduction reaction (CORR) and $NO_2^-$ reduction reaction ($NO_2^-$RR) performance of $Ru_1Cu$ SAA were firstly evaluated. As shown in Fig. 2a, under a pure CO gas feed, $Ru_1Cu$ SAA mainly produces hydrogen products and a small amount of acetate (Supplementary Fig. 14). In addition, $Ru_1Cu$ SAA exhibits high selectivity of $NH_3$ with a FE of -100% for $NO_2^-$RR (Fig. 2b), which might be attributed to the synergistic effect derived from SAA and optimization of the electronic structure[22,27,33]. Interestingly, when a CO-saturated aqueous solution containing 1 M KOH and 1 M $KNO_2$ was used as the electrolyte, the linear sweep voltammetry (LSV) curve of $Ru_1Cu$ SAA exhibits an enhanced current density under mixed feed gas compared to pure CORR (Supplementary Fig. 15), which indicates the occurrence of the electrocatalytic C−N coupling reaction. Notably, new NMR peaks appearing in the $^1H$ NMR spectrum of the electrolyte solution after electrolysis match the spectrum of formamide (Fig. 2c), which was not present in the sole CORR and $NO_2^-$RR. Furthermore, we further confirmed the production of formamide in testing the electrolyte solution by gas chromatography-mass spectrometry (GC-MS) (Fig. 2d). The experimental results show that formamide is only produced by the electrocatalytic co-reduction of CO and $NO_2^-$. In order to further clarify the sources of C and N in formamide, we carried out isotope labeling experiments using $^{15}N$-labeled $NO_2^-$ and $^{13}C$-labeled CO as raw materials. The $^1H$ NMR spectrum of the electrolyte after the coupling reaction of $^{15}NO_2^-$ and $^{13}CO$ shows typical $H^{13}CO^{15}NH_2$ peaks (Fig. 2c). Meanwhile, GC-MS measurements further confirmed the production of $H^{13}CO^{15}NH_2$ (Fig. 2d). These results demonstrate that the generated $HCONH_2$ originated from the electrocatalytic coupling reaction of CO and $NO_2^-$ on $Ru_1Cu$ SAA.

The Cu NCs, $Ru_1Cu$ SAA, and RuCu NPs were further adopted as catalysts for formamide electrosynthesis by simultaneously reducing CO and $NO_2^-$ (Supplementary Fig. 16). The Cu NCs shows high selectivity toward $NH_3$, along with a small amount of formamide (Supplementary Fig. 17a). In stark contrast, $Ru_1Cu$ SAA delivers higher formamide FE and yield at various reduction potentials, with a maximum formamide FE of $45.65 \pm 0.76\%$ and yield of $2483.77 \pm 155.34$ µg h$^{-1}$ mg$_{cat.}^{-1}$ at −0.5 V vs.

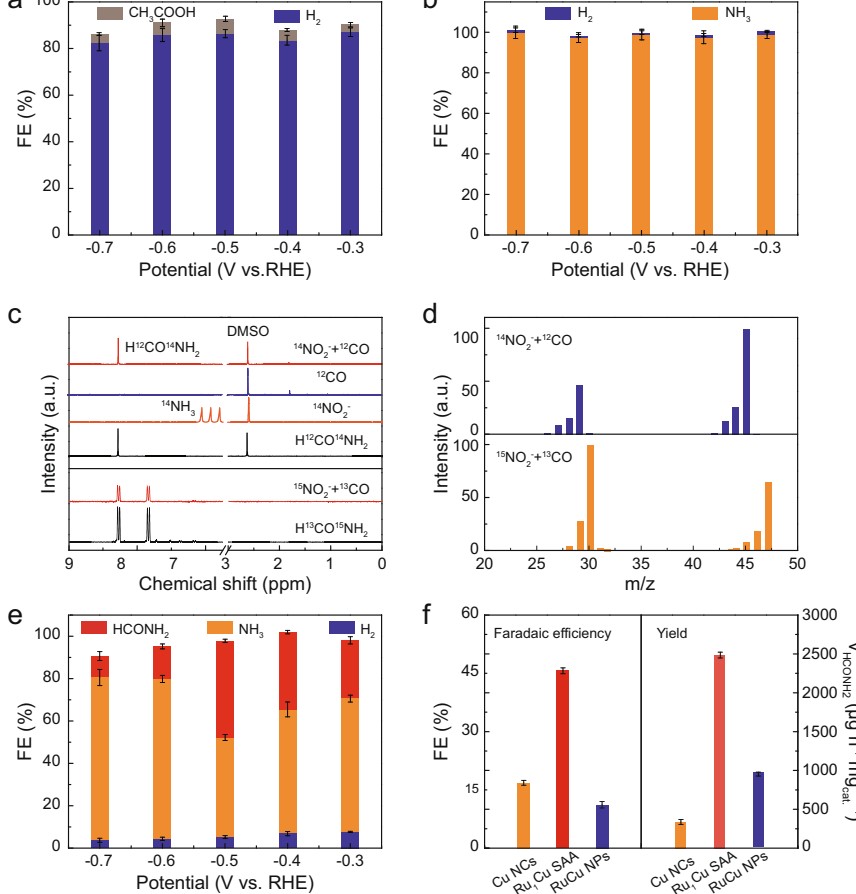

**Fig. 2 | Evaluation of electrocatalytic performance and qualitative detection of products. a, b** Faradaic efficiencies of major reduction products on Ru₁Cu SAA for **a** CORR and **b** NO₂⁻RR. **c** ¹H NMR spectra of standard references (HCONH₂ and H¹³CO¹⁵NH₂) and the electrolyte obtained after CORR, NO₂⁻RR, and NO₂⁻ + CO/¹⁵NO₂⁻ + ¹³CO co-reduction. **d** GC-MS results of the electrolyte obtained after NO₂⁻ + CO and ¹⁵NO₂⁻ + ¹³CO co-reduction. **e** Products distribution at different applied potentials in a CO-saturated 1 M KOH + 1 M KNO₂ solution on Ru₁Cu SAA. **f** Highest formamide Faradaic efficiencies and Yield of Cu NCs, Ru₁Cu SAA, and RuCu NPs. The error bars represent the standard deviation for at least three independent measurements.

RHE (Fig. 2e), respectively. When the Ru loading was further increased, the RuCu NPs exhibit higher hydrogen evolution reaction (HER) activity, thus resulting in lower the FE and yield of formamide (Fig. 2f and Supplementary Fig. 17b). These results suggest that the introduction of single-atom Ru can effectively improve the catalytic activity and selectivity of formamide synthesis while avoiding the promotion of competitive HER (Supplementary Fig. 18). To evaluate the intrinsic activities of the catalysts, we performed electrochemical double-layer capacitance ($C_{dl}$) measurements to normalize the electrochemically active surface area (ECSA) (Supplementary Fig. 19 and Supplementary Table 3)[27,34]. Although these catalysts present similar ECSA, Ru₁Cu SAA still shows the best intrinsic activity for C−N coupling towards formamide, indicating that the atomic dispersion of Ru in Ru₁Cu SAA enhances the intrinsic activity. Moreover, the durability of the Ru₁Cu SAA is evaluated by chronoamperometry in CO-saturated aqueous solution containing 1 M KOH and 1 M KNO₂ electrolyte. The current density, formamide FE, and yield of Ru₁Cu SAA show a negligible change at constant applied potential after 52 h of continuous electrolysis (Supplementary Fig. 20). Furthermore, we further investigated the stability of the Ru₁Cu SAA by using a membrane electrode assembly (MEA) electrolyzer. Remarkably, the Ru₁Cu SAA that could maintain full-cell voltage durability at high current density (-250 mA cm⁻²) for 50 h, as well as structural and chemical stability (Supplementary Fig. 21). These results strongly suggest that the Ru₁Cu SAA has excellent stability for formamide electrosynthesis.

## Investigation of formamide electrosynthesis mechanism

To elucidate the origins of the C−N coupling activity on the Ru₁Cu SAA catalyst, in situ XAS measurements using a homemade cell were initially performed to probe the electronic structure and local atomic environment changes of Ru₁Cu SAA during real electrosynthesis process[35]. During in situ XAS measurements, the applied potential was first increased from open circuit voltage (OCV) to −0.3 and −0.5 V vs. RHE, and then decreased back to OCV. Figure 3a shows the normalized operating Ru K-edge XANES at different applied potentials. Compared with the Ex situ condition, the absorption edge of the Ru K-edge XANES spectrum shifts toward the high-energy side (about 0.5 eV) (insert of Fig. 3a) under the OCV condition, while the white line peak broadens (Orange marked in Fig. 3a), implying an increase in the Ru oxidation state due to the binding of H₂O and NO₂⁻[36]. Furthermore, the absorption edge of Ru K-edge XANES shows a clear shift to the lower energies with increasing applied potential (inset of Fig. 3a), which is a combination of ligand effects and dissociation of the reactants or reaction intermediates. This probably results in Cu transfers electrons to Ru during C−N coupling, which in turn modulates the state of the adsorbed reactant intermediates and improves the catalytic activity of the alloy[23,36]. Meanwhile, the dissociation of reactants or reaction intermediates at Ru sites occurs and the recovery of the low oxidation state Ru reflects the rapid dissociation process of reactants or reaction intermediates on Ru atoms[37]. The results are further verified by corresponding FT-EXAFS spectra shown in Fig. 3b. In comparison with the

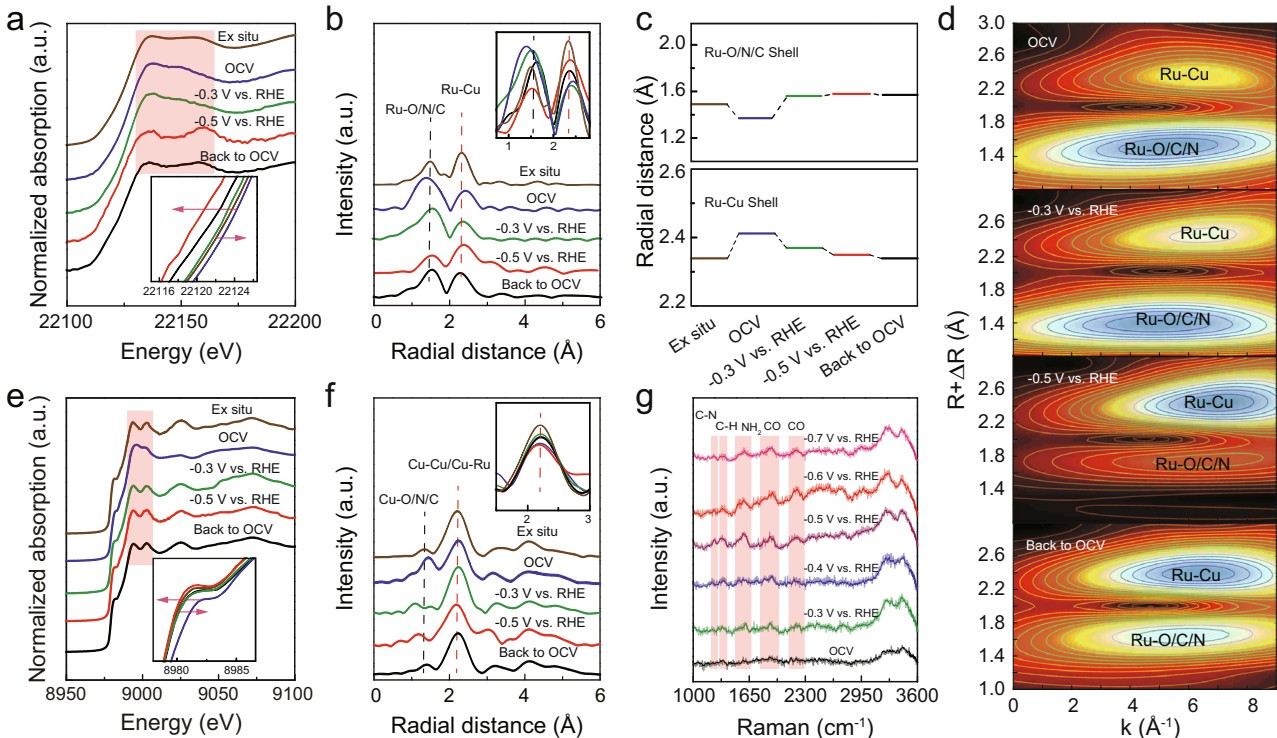

**Fig. 3 | In situ XAS and in situ Raman measurements under various applied potentials for Ru₁Cu SAA during electrocatalytic coupling of carbon monoxide and nitrite. a** In situ XANES spectra of Ru₁Cu SAA recorded at Ru K-edge. **b** Ru K-edge FT-EXAFS spectra for Ru₁Cu SAA. **c,** The variation of the radial distance of Ru-O/C/N and Ru-Cu shells at different applied potentials. **d** Corresponding WT contour profiles. **e** Cu K-edge XANES spectra for Ru₁Cu SAA. **f** Cu K-edge FT-EXAFS spectra for Ru₁Cu SAA. **g** In situ Raman spectra of Ru₁Cu SAA.

Ex situ condition, an enhanced scattering peak of the Ru-C/N/O shell obtained under OCV condition displays a negative shift (0.12 Å) (Fig. 3b and Fig. 3c), which is ascribed to the adsorption of reactants on Ru atoms for the generation of C–N coupling[38,39]. With the increase of the potential, the scattering intensity of Ru–O/N/C shell gradually decreases (insert of Fig. 3b), indicating that a large amount of reactants (CO or NO₂⁻) are consumed and oxidation state Ru reduction[40,41]. In addition, the scattering peaks of Ru–O/N/C shells gradually shift positively (Fig. 3c), which may be caused by the change of adsorbed species at Ru sites[42]. Meanwhile, the scattering peak of the Ru–Cu shell shows a low R shift and an enhanced intensity, which is caused by the reaction intermediates bound to Ru site and low oxidation state Ru recovery[43]. Moreover, the wavelet transforms (WT) spectrum of the corresponding Ru K-edge EXAFS oscillations further validates the above results (Fig. 3d). However, when the electrode potential was switched back to OCV, the Ru K-edges XANES, and FT-EXAFS spectra show signs of irreversible changes, possibly due to the strong adsorption of intermediate groups on Ru sites[44]. Figure 3e shows the in situ XANES spectra of Ru₁Cu SAA at Cu K-edge. This is a remarkable positive shift under OCV condition compared with that under Ex situ condition, indicating that the reactants are adsorbed on the Cu atoms, which is also evidenced by the emerging Cu-O/N/C shell scattering peaks in the corresponding FT-EXAFS spectra (Fig. 3f). Notably, when the potential is applied, the intensity of Cu–O/N/C shell scattering decreases sharply, indicating that the reactants are rapidly consumed at the Cu sites[40,41]. Moreover, compared to Ex situ conditions, the lower intensity of Cu–Cu/Cu–Ru scattering under OCV and applied potential conditions is observed on the Ru₁Cu SAA, especially the lowest intensity under −0.5 V vs. RHE conditions (inset of Fig. 3f), indicating the presence of a large amount of unsaturated coordination Cu under in situ conditions, which is favorable for the adsorption of reactants on Cu sites[45]. However, the majority of Ru₁Cu SAA are Cu atoms, whose

atomic structure is difficult to significantly change[25,46]. This results in Cu–Cu/Cu–Ru bond lengths with little variations during the entire in situ measurement (Fig. 3f). Thus, in situ XAS results give experimental evidence that C/N species adsorb and interact with surface Ru and Cu sites in Ru₁Cu SAA.

In situ Raman spectroscopy measurements were further conducted on Ru₁Cu SAA to validate the plausibility of the mechanism of carboxamide formation by distinguishing intermediates at the molecular level (Supplementary Fig. 22). Figure 3g exhibits the in situ Raman spectra of the Ru₁Cu SAA by utilizing an electrolyte containing CO-saturated 1 M KOH and 1 M KNO₂ at different operated potentials. When the applied potential is increased to −0.5 V vs. RHE, a notable peak corresponding to the stretching vibration of the C–N bond was observed on Ru₁Cu SAA[47]. Meanwhile, the C–H deformation at 1392 cm⁻¹, NH₂ at 1592 cm⁻¹, and CO stretching at 1890 and 2190 cm⁻¹ also started to appear[47–49]. These results further confirm the C–N bond formation and the occurrence of electrocatalytic processes. Moreover, the Raman signals of C–C, CH₂, and CH₃ stretching vibration were not observed, which indicates that the product of Ru₁Cu SAA C–N coupling reaction is formamide, not other organic compounds.

To distinguish the roles of the individual components of Ru₁Cu SAA in the C–N coupling process, we further investigated the formation of HCONH₂ by C–N coupling at the Ru₁Cu SAA interface by DFT calculations. The calculated adsorption energies of NO₂⁻ and CO on the optimized Ru₁Cu SAA model (Supplementary Fig. 23 and Supplementary Fig. S24) confirm that Ru₁Cu SAA is more favorable for the adsorption of NO₂⁻, suggesting that NO₂⁻ will be enriched on the surface of Ru₁Cu SAA. Moreover, Bader charge analysis determined that the adsorbed *NO₂ species obtained −0.94 |e| from Ru₁Cu SAA, while the adsorbed *CO species obtained −0.67 |e|. More charge transfer means that the interaction between Ru sites and *NO₂ species

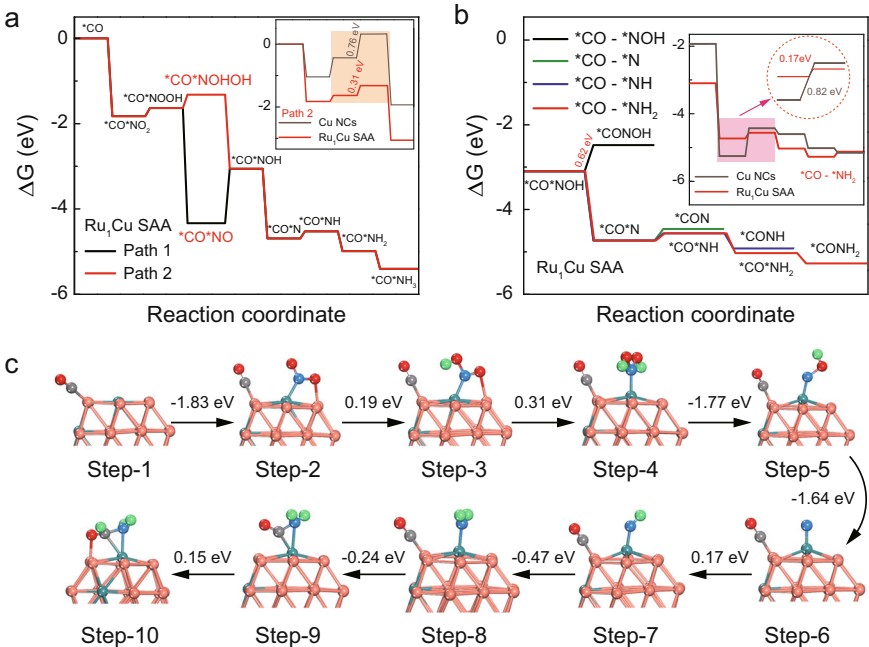

**Fig. 4 | Density functional theory (DFT) calculations. a** Diagram of free energy changes for $NO_2^-$ reduction on $Ru_1Cu$ SAA surface with the assistance of *CO. Insert shows free-energy diagram for $NO_2^-$RR on $Ru_1Cu$ SAA and Cu NCs. **b** Free energy diagrams of different reaction pathways for the synthesis of formamide on $Ru_1Cu$ SAA. Insert shows free-energy diagram for formamide production on $Ru_1Cu$ SAA and Cu NCs. **c** Diagram of the optimal energy pathway for the synthesis of formamide on $Ru_1Cu$ SAA and the corresponding atomic configurations for each step, Cu, Ru, C, O, N, and H atoms shown as orange, blue, gray, red, dark blue, and green, respectively.

is stronger than that of between Ru sites and CO* species (Supplementary Fig. 25). It can be inferred that the Ru sites are more favorable for the adsorption of $NO_2^-$ during the reaction process, while CO is more inclined to adsorb on the Cu sites close to the Ru sites. Moreover, when CO was adsorbed on Cu site adjacent to Ru, the Gibbs free energy of $NO_2^-$ on Ru site can be reduced from −1.80 eV (without *CO) to −1.83 eV (with *CO), indicating that the existence of activated CO molecules on Cu site can further promote the adsorption and activation of $NO_2^-$ on adjacent metal sites (Supplementary Fig. 26). Based on the above analysis, we proposed two possible reaction paths (with *CO) of $NO_2^-$RR (Fig. 4a and Supplementary Fig. 27). The results show that the rate-determining step (RDS) energy barrier of path 2 (*CO*NOOH → *CO*NOHOH, $\Delta G = 0.31$ eV) has a lower than that of path 1 (*CO*NO → *CO*NOH, $\Delta G = 1.28$ eV). Therefore, it can be inferred that the formation of *NOHOH intermediate instead of *NO can reduce the RDS energy barrier of the whole reaction, which is beneficial to accelerate the subsequent protonation process and enhance the electrosynthesis performance of formamide. Interestingly, the assistance of CO did not alter the $NO_2^-$RR reaction pathway, but lowered the energy barrier of the RDS (Supplementary Fig. 28). It can be inferred that CO adsorbed at the Cu site accelerates $NO_2^-$RR, which promotes the production of nitrogen-containing intermediates at the Ru site. Furthermore, the RDS of Cu NCs is *CO*N → *CO*NH ($\Delta G = 0.76$ eV), while that of $Ru_1Cu$ SAA is *CO*NOHOH → *CO*NH ($\Delta G = 0.31$ eV) (insert of Fig. 4a and Supplementary Fig. 29). On the other hand, the kinetic barriers for the formation of *CO*NOHOH and *CO*NH on $Ru_1Cu$ SAA are 0.67 and 0.91 eV respectively (Supplementary Fig. 30). However, the energy barriers are 1.05 and 1.03 eV for Cu NC, which is higher than $Ru_1Cu$ SAA. Indicating that the synthesis of formamide is thermodynamically and kinetically preferred in $Ru_1Cu$ SAA, which is in good agreement with our experimental observations. The results indicate that the introduction of highly dispersed Ru atoms can effectively improve the adsorption and activation of $NO_2^-$.

The calculation results suggest that various N-containing intermediates can be generated during the $NO_2^-$RR process, so the adsorbed CO species may be coupled with various nitrogen-containing intermediates, such as *NOH, *N, *NH, and *NH$_2$. Therefore, we have carried out a detailed study on the potential C−N coupling mechanism of electrosynthesis formamide. As shown in Fig. 4b, the coupling reaction of *CO with *NH$_2$ intermediates to form *CONH$_2$ ($\Delta G = -0.24$ eV) is thermodynamically the most favorable compared to *CO*NOH → *CONOH ($\Delta G = 0.62$ eV), *CO*N → *CO*NH ($\Delta G = 0.17$ eV) and *CO*NH → *CONH ($\Delta G = -0.36$ eV). The *CO*NH$_2$ → *CO*NH$_2$ process is an exothermic process, which indicates that the adjacent Ru−Cu dual sites can spontaneously couple *CO and *NH$_2$ intermediates to achieve the critical C−N coupling reaction. In addition, it is generally known that the C−C coupling reaction is known to be an energy-absorbing step[14,50], as is the coupling of carbon-containing intermediates with free NH$_3$[14], while the coupling of *CO with *NH$_2$ (*CO*NH$_2$ → *CO*NH$_2$, $\Delta G = -0.24$ eV) is an energy-releasing step. Therefore, the C−N coupling synthesis of formamide is more favorable in terms of thermodynamics and kinetics. This confirms the poor catalytic activity of $Ru_1Cu$ SAA in the synthesis of multi-carbon organics. Notably, the energy barrier of *CO*N → *CO*NH ($\Delta G$, 0.82 eV→0.17 eV) is significantly lower after the introduction of single atom Ru in the Cu nanoclusters (insert of Fig. 4b), suggesting that the introduction of highly dispersed Ru atoms can effectively promote $NO_2^-$ protonation and participate in the subsequent C−N coupling process. These results indicate that formamide generation via coupling of *CO and *NH$_2$ intermediates is thermodynamically and kinetically feasible. According to the above analysis, the overall reaction scheme can be best described as a suitable ten-step electrocatalytic cascade (Fig. 4c). With CO adsorption at Ru adjacent Cu sites, $NO_2^-$ preferentially undergoes deoxy-hydrogenation at Ru sites until the key *NH$_2$ intermediate is finally formed. Then, *CO adsorbed on Cu sites is spontaneously coupled with *NH$_2$ on Ru sites to generate a critical C−N bond, enabling formamide electrosynthesis.

## Discussion

In summary, we demonstrated a sustainable electrochemical approach to produce formamide through electrolysis co-reduction CO and $NO_2^-$ pollutants at ambient conditions. Highly selective formamide production with a Faradaic efficiency of $45.65 \pm 0.76\%$ and a yield of $2483.77 \pm 155.34\ \mu g\ h^{-1}\ mg_{cat.}^{-1}$ at $-0.5$ V vs. RHE are achieved on a Ru-dispersed Cu nanocluster electrocatalyst. As evidenced by in situ XAS, in situ Raman, and theoretical calculation, the key *CO and *NH₂ intermediates tend to spontaneously couple with low energy barriers at adjacent Ru−Cu dual active sites, leading to highly selective synthesis of formamides. This work opens a avenue for sustainable formamide electrosynthesis from cheap CO and $NO_2^-$ pollutant through the C−N coupling, providing a dual-sites design strategy based on single-atom alloy for the synthesis of various high-value-added chemicals.

## Methods

### Materials fabrication

The fabrication of $TiO_2$ nanowires is described elsewhere[21]. 40 mg $TiO_2$ nanowires was first dispersed in 10 mL of $H_2O$ with sonication to get well-dispersed $TiO_2$ nanowires suspension. Subsequently, 200 μL Copper (II) Chloride Dihydrate solution (Adamas, 98%) ($Cu^{2+}$: 10 mg mL$^{-1}$, 2 mg) was added to the prepared $TiO_2$ nanowires aqueous solution with stirring for 60 min at room temperature. The mixed solution was frozen in a liquid nitrogen environment and then freeze-dried. Then, $TiO_2$ nanowires anchored with Cu NCs were obtained via a reduction treatment of the freeze-dried samples in a $H_2/Ar$ ($H_2$: Ar = 20: 180 sccm) stream at 300 °C (heating rate: 5 °C min$^{-1}$) for 2 h. Finally, the supported $Ru_1Cu$ SAA samples were prepared via a galvanic replacement method. The $TiO_2$ nanowires anchored with Cu NCs sample was dispersed in deionized water (10 mL), followed by dropwise adding desired amount of $RuCl_3$ solution (Adamas, 99.5%⁺, Ru: 47%) ($Ru^{3+}$: 10 mg mL$^{-1}$, 0.15 mg) in an ultrasonic water bath. The obtained slurry was centrifuged and washed with distilled water, and then dried. As control samples, RuCu NPs were prepared by the same method (The mass ratios of the precursor materials are $TiO_2$: Cu: Ru = 40:3:1, respectively).

### Structural characterization

XRD patterns of the samples were taken by using a Rigaku MiniFlex X-ray diffraction. The surface topography of the samples was characterized with a Tescan MIRA3 SEM, equipped with an Oxford energy-dispersive X-ray spectroscope. The transmission electron microscopy (TEM), high-angle annular dark field-scanning TEM (HAADF-STEM), and element mapping were taken by a Thermo scientific Themis Z (3.2) with double spherical aberration (Cs) correctors for both the probe-forming. The chemical state and composition of the samples were characterized using X-ray photoelectron spectroscopy (XPS, Thermo Scientific Escalab 250Xi). X-ray absorption near-edge structure (XANES) and Fourier transform (FT) curve of extended X-ray absorption fine structure (EXAFS) spectra were measured at beamline BL01C1 of Taiwan light source. The contents of Ru and Cu were obtained via the inductively coupled plasma-optical emission spectrometry (ICP-OES) (Agilent 730). In situ Raman spectroscopy was performed by a WITec Alpha300R (WITec GmbH, Germany) confocal spectrometer equipped with a 532 nm single longitudinal-mode laser at room temperature.

### Preparation of the electrode

To prepare the catalyst ink, the catalyst was ultrasonically dispersed in solution (40 μL of Nafion solution (Adamas, RG, 5 wt%), 960 μL of absolute ethanol), and ultrasonicated for 30 min to form a uniform ink. The homogeneous ink was loaded onto the gas diffusion layer (Sigraset 29 BC) electrodes and dried under ambient conditions. The catalyst loading was estimated to be ~0.6 mg cm$^{-2}$.

### Electrochemical formamide synthesis

The electrochemical experiments were performed on an electrochemical workstation (CHI660E) using an H-Cell with a three-electrode configuration (working electrode, Pt plate counter electrode, and Hg/HgO/saturated 1 M KOH reference electrode). the cathode and anode chambers were separated by an anion exchange membrane (FAB-PK-130). All of the potential measurements were converted to the RHE using the following formula: $E_{RHE} = E_{Hg/HgO} + 0.097 + 0.0591 \times pH$.

For electrochemical formamide synthesis, potentiostatic tests were carried out in CO-saturated 1 M $KNO_2$ + 1 M KOH (CO, Changsha Gaoke Gas Co., 99.9999%; $KNO_2$, Adamas, RG; KOH, Greagent, AR), which was bubbled with CO for 20 min before the measurement. During the experiment, CO enters the cathode chamber continuously at a constant flow rate.

For the stability tests in MEA electrolyzers, the experimental set-up used was a commercial MEA electrolyzer (4 cm²). The MEA consisted of a cathode electrode, anion-exchange membrane (FAB-PK-130), and anode electrode ($IrO_2$-Ti mesh).

### Electrochemical CORR measurements

For electrochemical CORR tests, potentiostatic tests were carried out in 1 M KOH without $NO_2^-$.

### Electrochemical $NO_2^-$RR measurements

For electrochemical $NO_2^-$RR tests, potentiostatic tests were carried out in 1 M $KNO_2$ + 1 M KOH without feeding CO.

### Product qualitative and quantification

A gas chromatograph equipped with a flame ionization detector (FID) and a thermal conductivity detector (TCD) was used for the quantification of the gaseous products. The gas chromatography used high purity argon (99.999%) as carrier gas. Organic liquid products were quantified by Bruker 400 MHz NMR spectrometer. The NMR samples were prepared by mixing 0.5 mL of electrolyte with 0.1 mL of deuterated water ($D_2O$), and 0.02 μL of dimethyl sulfoxide (DMSO) was added as an internal standard. Unusually, the test solution of $NO_2^-$RR was acidified (pH adjusted to ~2) before NMR testing. In addition, the production of formamide was further verified by GC-MS (gas chromatography-mass spectrometry) (Agilent 59771A).

### Determination of $NH_3$

After diluting the post-test electrolyte solution to the appropriate concentration, the $NH_3$ concentration in the electrolyte solution was detected spectrophotometrically modified by the modified indophenol blue method[51]. 2 mL of the diluted electrolyte solution was taken and 2 mL of NaOH (1 M) solution containing salicylic acid (5 wt%) (99%, Adamas) and sodium citrate dihydrate (5 wt%) (AR, Greagent), 1 mL of 0.05 M NaClO (Active chlorine ≥7.5%: Greagent), and 0.2 mL of a 1 wt% $C_5FeN_6Na_2O$ (sodium nitroferricyanide, 99%⁺, Adamas) aqueous solution were added sequentially. After being left at room temperature for 2 h, the absorption spectra of the developed solutions were detected by UV-Vis spectrophotometer (Shimadzu, UV-2600). The formation of indophenol blue was determined by absorbance at a fixed wavelength of 655 nm. The concentration-absorbance curves were calibrated using standard ammonia sulfate solutions (($NH_4)_2SO_4$, Greagent, ≥99.5%), as shown in Supplementary Fig. 13.

### Isotope labeling experiments

Isotope labeling experiments were performed using $Na^{15}NO_2$ (99 at.% of $^{15}N$, Shanghai Macklin Biochemical Technology Co., Ltd.) as the N source and $^{13}CO$ (99 at.% of $^{13}C$, Wuhan Newradar Special Gas Co. Ltd) as the C source. After potentiostatic electrolysis at $-0.5$ V (vs. RHE), the catholyte was collected and concentrated for NMR and GC-MS analysis.

## Calculation of Faradaic efficiency for corresponding product

The Faradaic efficiency of each gas product was calculated by the equation:

$$\text{Faradaic efficiency (\%)} = (nFxV)/j * 100 \qquad (1)$$

where $n$ is the number of electrons transferred, $F$ is Faraday's constant, $x$ is the mole fraction of product, $V$ is the total molar flow rate of gas and $j$ is the total current.

The Faradaic efficiency for liquid products generation was calculated as follows:

$$\text{Faradaic efficiency (\%)} = (nFCV)/Q * 100 \qquad (2)$$

Where $n$ is the number of electrons transferred; $V$ is electrolyte volume; $C$ is the concentration of liquid products; $F$ is Faraday's constant; $Q$ is the electric quantity.

## Calculation of yield rate for corresponding product

The average yield rate was calculated as follows:

$$v = (cV)/(tm) \qquad (3)$$

where $c_{formamide}$ is corresponding product concentration ($\mu g\,mL^{-1}$), $V$ is the total volume of electrolyte (mL), $t$ is time (h) for electrocatalysis and $m$ is the catalyst loadings (mg).

## In situ XAS measurements

The corresponding in situ XAS spectroscopic measurements (Ru K- and Cu K-edges XAS) were performed at the BL01C1 beamline at the National Synchrotron Radiation Research Center (NSRRC, Taiwan). The in situ XAS measurements were performed in a customized three-electrode cell with a carbon rod and a saturated glycury electrode (SCE) as the counter and reference electrode, respectively. Catalyst ink was dropped on a carbon cloth substrate as the working electrode. Potentiostatic tests were performed in a CO-saturated 1 M $KNO_2$ + 1 M KOH electrolyte with CO bubbling for 20 min before measurement. The working electrode was covered with Kapton film on one side facing the incident X-rays, while the other side was in contact with the electrolyte. XAS spectra were measured in fluorescence mode at room temperature. The obtained XAS data were processed with the ATHENA program.

## Electrochemical in situ Raman measurements

The in situ Raman measurement tests were performed using a customized three-electrode cell with a platinum wire and an Ag/AgCl electrode as the counter and reference electrode, respectively. The catalyst ink was dropped onto a glassy carbon substrate as a working electrode. In situ Raman spectra were recorded using a Thermo Fisher DXR2 Raman microscope with DXR 532 nm laser as the excitation source.

## Computational method

All the density functional theory (DFT) calculations were performed by using the Vienna ab initio Simulation Package (VASP) with the projector augmented wave (PAW) potentials[52,53]. The generalized gradient approximation (GGA)/Perdew-Burke-Ernzerhof (PBE) level were adopted[54]. All the atomic positions were allowed to relax until the forces were less than 0.02 eV/Å, and the electron convergence energy was set to $10^{-5}$ eV. The cutoff was set to 400 eV to expand wave function. The boxes were set to $15 \times 15 \times 15$ for all the $Ru_1Cu$ SAA structures.

In order to simplify the model, we adopted the $Cu_{38}$ cluster model as the candidate catalyst substrate, where the $Ru_1Cu$ SAA ($Cu_{36}Ru_2$) can be achieved by replacing two Cu atoms with two Ru atoms. To screen out the stable $Ru_1Cu$ SAA structure, the formation energies can be

considered as follows:

$$E_f = E_{Cu_{36}Ru_2} + 2E_{Cu_1} - E_{Cu_{38}} - 2E_{Ru_1} \qquad (4)$$

where the $E_{Cu_{36}Ru_2}$, $E_{Cu_1}$, $E_{Cu_{38}}$, and $E_{Ru_1}$ are the energies for $Ru_1Cu$ SAA, single Cu atom, $Cu_{38}$ cluster, and single Ru atom, respectively.

In this work, the adsorption energies of $NO_2^-$ and/or CO on $Ru_1Cu$ SAA interface were calculated, respectively, can be defined as follows:

$$E_{ads} = E_{Cu_{36}Ru_2 + NO_2/CO} - E_{Cu_{38}Ru_2} - E_{NO_2/CO} \qquad (5)$$

where the $E_{Cu_{36}Ru_2 + NO_2/CO}$, $E_{Cu_{36}Ru_2}$, and $E_{NO_2/CO}$ are the energies for $NO_2^-$ and/or CO adsorption on $Ru_1Cu$ SAA interface, $Ru_1Cu$ SAA, $NO_2^-$ or CO in gas, respectively.

## Data availability

The data supporting the findings of this study are available from the corresponding authors upon reasonable request.

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

## Acknowledgements

This work was supported by the National Natural Science Foundation of China (no. 51771072 (Y.W.T.)), the Youth 1000 Talent Program of China (Y.W.T.), the Outstanding Youth Scientist Foundation of Hunan Province (no. 2020JJ2006 (Y.W.T.)), the Fundamental Research Funds for the Central Universities, Hunan University State Key Laboratory of Advanced Design and Manufacturing for Vehicle Body Independent Research Project (no. 71860007), and Hunan Provincial Innovation Foundation for Postgraduate (no. CX20220415 (J.L.)).

## Author contributions

Y.W.T. conceived and directed the project. J.L. carried out key experiments. Z.X.W. and S.L.Z. performed theoretical calculations. J.L., D.C.C., Y.R.L., and T.S.C. contributed to the XAS measurements and analyses of the XAS experiment results. Y.W.T. and J.L. wrote the manuscript with input from all other authors. All authors discussed the results and commented on the manuscript.

## Competing interests

The authors declare no competing interests.
