## [Peer review file · Nature Communications]

REVIEWER COMMENTS

Reviewer #1 (Remarks to the Author):

The manuscript investigates the RuCu single-atom alloy (SAA) formations in electrochemical formamide synthesis. The authors provided detailed structural analysis and performed DFT simulations to understand the role of Ru-Cu bonding on the electrochemical reaction. The manuscript investigates the bifunctional role of RuCu SAAs in formamide synthesis, which would interest the scientific community. However, the manuscript lacks a detailed explanation of EXAFS analysis, does not provide an optimized EXAFS model, and does not explain the EXAFS spectra in depth along with the DFT simulations. The manuscript could potentially be impactful if the operando EXAFS spectra and DFT simulations are used to describe the reaction mechanism and structural transformation during the reaction. However, this is not done in the manuscript. DFT simulations proposed a reaction pathway, yet binding energy calculations do not provide evidence that the proposed reaction coordinate system would be valid. Transition state and reaction pathway analyses were not provided. Therefore, although the manuscript has the potential to be an impactful work, EXAFS analysis and DFT analysis are not complete. The manuscript requires major revisions. The additional comments are stated below.

- The authors state that energy consumption of the industrial synthesis of the formamide process is high. To understand the impact of this work, the energy efficiency gained in the new catalyst reported in the manuscript must be reported, and energy and conversion efficiency should be compared with the industrial catalyst.
- The Ru and Cu precursor concentrations and expected wt% should be reported in the synthesis and compared with the actual concentrations.
- The Ru 3d 5/2 and 3/2 peak ratios in Fig S5 do not seem well aligned with the theoretical ratio. What is the ratio? This should be clarified. XPS data of Ru metal and Ru oxide would be helpful to explain the 0.54 eV shift. The shifts should be explained considering the oxidation state of Ru.
- The manuscript must provide a detailed model of the EXAFS spectra. This is vital to compare the DFT predictions and accuracy of the DFT modeling structures, as well as to give a confident analysis of the EXAFS analysis. The operando EXAFS analysis should also be reanalyzed once the proper model is proposed.
- The authors stated that a primitive model and single path fitting are provided in Fig S7 and Table S2. How the EXAFS model was chosen is unclear, and Table S2 is confusing. The first and second shell scattering paths should be provided.
- The comparison of the DFT model with the EXAFS analysis must be provided in the manuscript. The proposed DFT structures should be used to model the EXAFS spectra.
- The error bars for the EXAFS fitting must be provided.
- When properly fitted, the reviewer recommends showing the results in the manuscript, not the supporting information, which is essential data that will combine the EXAFS and DFT analysis.
- The EXAFS analysis of RuCu nanoparticles should be provided.
- The Ru-O bonds observed in the EXAFS spectra are not considered in the DFT model. The effect of the Ru-O oxygen bond, evidenced in the EXAFS, questions the accuracy of the DFT structure.
- The STEM EDX image shows that the Ru clusters are forming. Most likely, the structures combine RuCu SAAs and Ru metal clusters. This can be normal. What is the evidence that Ru metals are not forming in these structures? This must be explained thoroughly in the manuscript.
- The authors state, "It is noteworthy that these isolated Ru atoms are surrounded by Cu atoms in different regions of the TiO₂ nanowires, rather than Ru nanoclusters." It is visible that Ru nanoclusters are also forming. This should be clarified in the manuscript.
- The reaction coordinate system is determined through the binding energies in DFT simulations. The rate-limiting steps are also determined by the energy difference. This is not necessarily an objective approach. The transition states were not calculated. Activation energies are unknown. Therefore, the proposed reaction mechanism must be justified.

Reviewer #2 (Remarks to the Author):

Lan et al. report the synthesis of a Ru₁Cu single-atom alloy electrocatalyst and its application for converting CO and nitrite into formamide. High Faradaic efficiencies up to 46% could be achieved for formamide production. Both operando XAS and in situ Raman measurements reveal the electrocatalytic processes of the C-N bond formation on catalyst surface. DFT calculations suggest that CO adsorption at Ru adjacent Cu sites and nitrite reduction at Ru sites can generate the C-N bond. The electroreduction of CO and nitrite to formamide is interesting, and it has not been reported in literature. However, there are quite a few technical problems, which should be addressed before considering publication in the journal.

- 1) The equations for calculating the Faradaic efficiency are problematic. The authors should also provide the detailed data to calculate FE of formamide.
- 2) TiO₂ nanowires were used as the support for the catalyst, why? How about the conductivity of TiO₂ nanowires?
- 3) The electron microscopy characterizations need improvement. The SEM image in Fig 1b is not clear. Figs 1c and 1d could not show clearly single Ru atom. Ru mapping in Fig 1e does not match well with the Cu mapping.
- 4) Some control reactions, for example the coreduction of: CO+nitrate, CO+NO, CO+hydroxylamine, CO+amine, should be conducted to better understand the C-N coupling process.
- 5) The current density in the stability test (Supplementary Fig. 19, -10 mA/cm²) is much smaller than the LSV measurement (Supplementary Fig. 14, -20 mA/cm² at -0.5 V), why?

Reviewer #3 (Remarks to the Author):

Title: Efficient Electrosynthesis of Formamide from Carbon Monoxide and Nitrite on a Ru-dispersed Cu nanocluster catalyst

This manuscript reported the electrochemical process for the synthesis of high-valued formamide from carbon monoxide and nitrite with a Ru₁Cu single-atom alloy under ambient conditions. Operando X-ray absorption spectroscopy, in situ Raman spectroscopy and density functional theory calculations results were used to detect intermediates. However, there are still some problems in the material characterization and properties investigation. Especially, the reaction mechanism is not clear.

1. Figure 1e is too fuzzy to distinguish Cu from Ru. The Electron energy loss spectroscopy (EELS) is better to distinguish Cu and Ru than HAADF-STEM image and the corresponding elemental mapping.
2. The authors adopted Galvanic replacement strategy to prepare Ru₁Cu single-atom alloy. What is the driving force for the formation of Ru₁Cu single-atom alloy? The Ru₁Cu single-atom structure is uniform? If not, how the authors chose the model for simulation?
3. As the authors say that the electrochemical synthesis of organic amides from C-N coupling reaction by introducing an NH₃ source during CO₂/CO reduction reaction, the formamide detected in this work may be formed by the reaction of NH₃ and CO. The labeled nitrite and unlabeled ammonia can be used as N-source to explore the origin of N under reductive conditions.
4. As shown in Figure 2a, the main C-related product is CH₃COOH (FE>80%), why is acetamide not detected in this work?
5. In Figure 2c, why is the H₂O peak of NO₂-+CO in 1H NMR a downward inverted peak?
6. In Figure 3g, the baseline of the -0.5 V spectrum is very uneven. The formamide has been formed in all potentials, why the intermediates can be only detected at -0.5 V? Isotope labeling experiments must be provided to better confirm the intermediates.
7. Stability is an important parameter to evaluate the materials. The authors should carry out the long-term test under high current density. In addition, the characterization of the catalytic materials after test should be provided.
8. The author should carefully check the horizontal coordinate of Figure 4a and should not make this

kind of mistake.

9. The resolution of all the pictures in the article is too low, which affects the reader's reading.

Reviewer #4 (Remarks to the Author):

In this paper, the authors prepare atomically dispersed Ru atoms on Cu nanoclusters single atom alloy loaded on TiO₂ nanowires. By using this catalyst, they conduct electroreduction of CO and NO₂⁻ to product formamide with high Faradaic efficiency. This work offers a new insight into the synthesis of organic nitrogen compounds under ambient conditions. However, several issues should be addressed before I can recommend publication. The following suggestions are provided for the authors:

1. TiO₂, as a widely used semiconductor in photocatalysis. Why the authors choose TiO₂ as the catalyst support?
2. In lines 204-208, "a remarkable positive shift under OCV condition compared with that under EX situ condition" and "the emerging Cu-O/N/C shell scattering peaks in the corresponding FT-EXAFS spectra" are ascribed to the adsorbed reactants on the Cu atoms. Authors need to provide more convincing evidences.
3. In line 224, "CO stretching at 1890 and 2903 cm⁻¹", this value in the text was mismatch with the results in Fig. 3g.
4. In Supplementary Fig. 20, XRD patten of Ru₁Cu SAA before stability test should be provided for better comparison.
5. As important control samples- RuCu NPs, the authors need to supply more basic characterizations such as SEM, HAADF-STEM.
6. Does the nitrite concentrate affected on the C-N coupling process?
7. The usage of "Operando" should be strict.

Minor suggestion: Some bottom of the words in some figures are covered, such as Fig. 2, Supplementary Fig. 6 and Supplementary Fig. 11b.

REVIEWER COMMENTS

Reviewer #1 (Remarks to the Author):

The manuscript investigates the RuCu single-atom alloy (SAA) formations in electrochemical formamide synthesis. The authors provided detailed structural analysis and performed DFT simulations to understand the role of Ru-Cu bonding on the electrochemical reaction. The manuscript investigates the bifunctional role of RuCu SAAs in formamide synthesis, which would interest the scientific community. However, the manuscript lacks a detailed explanation of EXAFS analysis, does not provide an optimized EXAFS model, and does not explain the EXAFS spectra in depth along with the DFT simulations. The manuscript could potentially be impactful if the operando EXAFS spectra and DFT simulations are used to describe the reaction mechanism and structural transformation during the reaction. However, this is not done in the manuscript. DFT simulations proposed a reaction pathway, yet binding energy calculations do not provide evidence that the proposed reaction coordinate system would be valid. Transition state and reaction pathway analyses were not provided. Therefore, although the manuscript has the potential to be an impactful work, EXAFS analysis and DFT analysis are not complete. The manuscript requires major revisions. The additional comments are stated below.

- The authors state that energy consumption of the industrial synthesis of the formamide process is high. To understand the impact of this work, the energy efficiency gained in the new catalyst reported in the manuscript must be reported, and energy and conversion efficiency should be compared with the industrial catalyst.
- The Ru and Cu precursor concentrations and expected wt% should be reported in the synthesis and compared with the actual concentrations.
- The Ru 3d 5/2 and 3/2 peak ratios in Fig S5 do not seem well aligned with the theoretical ratio. What is the ratio? This should be clarified. XPS data of Ru metal and Ru oxide would be helpful to explain the 0.54 eV shift. The shifts should be explained considering the oxidation state of Ru.
- The manuscript must provide a detailed model of the EXAFS spectra. This is vital to compare the DFT predictions and accuracy of the DFT modeling structures, as well as to give a confident analysis of the EXAFS analysis. The operando EXAFS analysis should also be reanalyzed once the proper model is proposed.
- The authors stated that a primitive model and single path fitting are provided in Fig S7 and Table S2. How the EXAFS model was chosen is unclear, and Table S2 is confusing. The first and second shell scattering paths should be provided.
- The comparison of the DFT model with the EXAFS analysis must be provided in the manuscript. The proposed DFT structures should be used to model the EXAFS spectra.
- The error bars for the EXAFS fitting must be provided.
- When properly fitted, the reviewer recommends showing the results in the manuscript, not the supporting information, which is essential data that will combine the EXAFS and DFT analysis.
- The EXAFS analysis of RuCu nanoparticles should be provided.

- The Ru-O bonds observed in the EXAFS spectra are not considered in the DFT model. The effect of the Ru-O oxygen bond, evidenced in the EXAFS, questions the accuracy of the DFT structure.
- The STEM EDX image shows that the Ru clusters are forming. Most likely, the structures combine RuCu SAAs and Ru metal clusters. This can be normal. What is the evidence that Ru metals are not forming in these structures? This must be explained thoroughly in the manuscript.
- The authors state, “It is noteworthy that these isolated Ru atoms are surrounded by Cu atoms in different regions of the TiO₂ nanowires, rather than Ru nanoclusters.” It is visible that Ru nanoclusters are also forming. This should be clarified in the manuscript.
- The reaction coordinate system is determined through the binding energies in DFT simulations. The rate-limiting steps are also determined by the energy difference. This is not necessarily an objective approach. The transition states were not calculated. Activation energies are unknown. Therefore, the proposed reaction mechanism must be justified.

Reviewer #2 (Remarks to the Author):

Lan et al. report the synthesis of a Ru₁Cu single-atom alloy electrocatalyst and its application for converting CO and nitrite into formamide. High Faradaic efficiencies up to 46% could be achieved for formamide production. Both operando XAS and in situ Raman measurements reveal the electrocatalytic processes of the C-N bond formation on catalyst surface. DFT calculations suggest that CO adsorption at Ru adjacent Cu sites and nitrite reduction at Ru sites can generate the C-N bond. The electroreduction of CO and nitrite to formamide is interesting, and it has not been reported in literature. However, there are quite a few technical problems, which should be addressed before considering publication in the journal.

- 1) The equations for calculating the Faradaic efficiency are problematic. The authors should also provide the detailed data to calculate FE of formamide.
- 2) TiO₂ nanowires were used as the support for the catalyst, why? How about the conductivity of TiO₂ nanowires?
- 3) The electron microscopy characterizations need improvement. The SEM image in Fig 1b is not clear. Fig 1c and 1d could not show clearly single Ru atom. Ru mapping in Fig 1e does not match well with the Cu mapping.
- 4) Some control reactions, for example the coreduction of: CO+nitrate, CO+NO, CO+hydroxylamine, CO+amine, should be conducted to better understand the C-N coupling process.
- 5) The current density in the stability test (Supplementary Fig. 19, -10 mA/cm²) is much smaller than the LSV measurement (Supplementary Fig. 14, -20 mA/cm² at -0.5 V), why?

Reviewer #3 (Remarks to the Author):

Title: Efficient Electrosynthesis of Formamide from Carbon Monoxide and Nitrite on a Ru-dispersed Cu nanocluster catalyst

This manuscript reported the electrochemical process for the synthesis of high-valued formamide from carbon monoxide and nitrite with a Ru₁Cu single-atom alloy under ambient conditions. Operando X-ray absorption spectroscopy, in situ Raman spectroscopy and density functional theory calculations results were used to detect intermediates. However, there are still some problems in the material characterization and properties investigation. Especially, the reaction mechanism is not clear.

1. Figure 1e is too fuzzy to distinguish Cu from Ru. The Electron energy loss spectroscopy (EELS) is better to distinguish Cu and Ru than HAADF-STEM image and the corresponding elemental mapping.

2. The authors adopted Galvanic replacement strategy to prepare Ru₁Cu single-atom alloy. What is the driving force for the formation of Ru₁Cu single-atom alloy? The Ru₁Cu single-atom structure is uniform? If not, how the authors chose the model for simulation?

3. As the authors say that the electrochemical synthesis of organic amides from C-N coupling reaction by introducing an NH₃ source during CO₂/CO reduction reaction, the formamide detected in this work may be formed by the reaction of NH₃ and CO. The labeled nitrite and unlabeled ammonia can be used as N-source to explore the origin of N under reductive conditions.

4. As shown in Figure 2a, the main C-related product is CH₃COOH (FE>80%), why is acetamide not detected in this work?

5. In Figure 2c, why is the H₂O peak of NO₂⁻+CO in ¹H NMR a downward inverted peak?

6. In Figure 3g, the baseline of the -0.5 V spectrum is very uneven. The formamide has been formed in all potentials, why the intermediates can be only detected at -0.5 V? Isotope labeling experiments must be provided to better confirm the intermediates.

7. Stability is an important parameter to evaluate the materials. The authors should carry out the long-term test under high current density. In addition, the characterization of the catalytic materials after test should be provided.

8. The author should carefully check the horizontal coordinate of Figure 4a and should not make this kind of mistake.

9. The resolution of all the pictures in the article is too low, which affects the reader's reading.

Reviewer #4 (Remarks to the Author):

In this paper, the authors prepare atomically dispersed Ru atoms on Cu nanoclusters single atom alloy loaded on TiO₂ nanowires. By using this catalyst, they conduct electroreduction of CO and NO₂⁻ to product formamide with high Faradaic efficiency. This work offers a new insight into the synthesis of organic nitrogen compounds under ambient conditions. However, several issues should be addressed before I can recommend publication. The following suggestions are provided for the authors:

1. TiO₂, as a widely used semiconductor in photocatalysis. Why the authors choose TiO₂ as the catalyst support?

2. In lines 204-208, “a remarkable positive shift under OCV condition compared with that under EX situ condition” and “the emerging Cu-O/N/C shell scattering peaks in the corresponding FT-EXAFS spectra” are ascribed to the adsorbed reactants on the Cu atoms. Authors need to provide more convincing evidences.
3. In line 224, “CO stretching at 1890 and 2903 cm^{-1} ”, this value in the text was mismatch with the results in Fig. 3g.
4. In Supplementary Fig. 20, XRD patten of Ru1Cu SAA before stability test should be provided for better comparison.
5. As important control samples- RuCu NPs, the authors need to supply more basic characterizations such as SEM, HAADF-STEM.
6. Does the nitrite concentrate affected on the C-N coupling process?
7. The usage of “Operando” should be strict.

Minor suggestion: Some bottom of the words in some figures are covered, such as Fig. 2, Supplementary Fig. 6 and Supplementary Fig. 11b.

Responses to the Referees' Comments

We would like to thank all the referees for the careful review and the valuable comments.

We have carefully considered the referees' comments and revised the manuscript accordingly. Below we list the changes we have made in light of the referees' comments.

Reviewer #1 (Remarks to the Author):

The manuscript investigates the RuCu single-atom alloy (SAA) formations in electrochemical formamide synthesis. The authors provided detailed structural analysis and performed DFT simulations to understand the role of Ru-Cu bonding on the electrochemical reaction. The manuscript investigates the bifunctional role of RuCu SAAs in formamide synthesis, which would interest the scientific community. However, the manuscript lacks a detailed explanation of EXAFS analysis, does not provide an optimized EXAFS model, and does not explain the EXAFS spectra in depth along with the DFT simulations. The manuscript could potentially be impactful if the operando EXAFS spectra and DFT simulations are used to describe the reaction mechanism and structural transformation during the reaction. However, this is not done in the manuscript. DFT simulations proposed a reaction pathway, yet binding energy calculations do not provide evidence that the proposed reaction coordinate system would be valid. Transition state and reaction pathway analyses were not provided. Therefore, although the manuscript has the potential to be an impactful work, EXAFS analysis and DFT analysis are not complete. The manuscript requires major revisions. The additional comments are stated below.

Reply: We appreciate the reviewer for recognizing of our work. We also thank the

reviewer for the deep and professional comments and suggestions, which are very valuable for improving the scientific impact of this work. By following the reviewer's comments and suggestions, we have conducted additional experiments and analyses. The details will be described below.

1. The authors state that energy consumption of the industrial synthesis of the formamide process is high. To understand the impact of this work, the energy efficiency gained in the new catalyst reported in the manuscript must be reported, and energy and conversion efficiency should be compared with the industrial catalyst.

Reply: We thank for the reviewer's valuable comment. The industrially used one-step synthesis of formamide ($\text{CO} + \text{NH}_3 \rightarrow \text{HCONH}_2$) from carbon monoxide and ammonia is directly synthesized under high pressure (10-30 MPa) and temperature 80-100°C under the catalysis of sodium methoxide (*Chem. Ing. Tech.* **62**, 434-434 (1990), *Nat. Commun.* **13**, 5452 (2022), *Chem. Soc. Rev.* **52**, 2193-2237 (2023)). In contrast, the synthesis method reported in this paper can be carried out under normal temperature and pressure conditions, and the reaction conditions are mild. Unfortunately, in the literature we have consulted so far, there is no report related to the energy efficiency of industrial synthesis of formamide. In addition, the accurate thermodynamic equilibrium potential parameters of formamide synthesis have not been obtained from the literature information, which is an obstacle to obtaining accurate energy conversion efficiency. Therefore, we are unable to precisely give an energy conversion efficiency for formamide synthesis. Thank you very much for your understanding. Moreover, the electrochemical systems for C-N coupling reactions are not yet well developed, so there

is not yet an extensive technoeconomic analysis (*Trends Chem.* **2**,1004-1019 (2020), *Nat. Rev. Chem.* **6**, 303-319 (2022)). However, a reasonable reference could be the analyses performed for electrochemical nitrogen reduction reactions and electrochemical CO₂ reduction reactions, which state that current densities in the range of hundreds of mA cm⁻² and selectivities of >50% need to be reached for a system to realize economic feasibility. By comparison, we found that the experimental results we obtained are still far from the goal of economic feasibility. Therefore, we will take it as a goal for further in-depth research (catalyst optimization, reactor design, etc.).

2. The Ru and Cu precursor concentrations and expected wt% should be reported in the synthesis and compared with the actual concentrations.

Reply: We are very grateful for the kind reminder from the reviewer. For this comment, we have added a detailed description of the concentrations of Ru and Cu precursors during material synthesis in the materials fabrication of the revised manuscript (page 14, line 20-22 and page 15, line 1-10). Moreover, we added relevant data as shown in **Table R1 (Supplementary Table 1)** in the revised manuscript). The results show that the Cu concentrations of Cu NCs and the actual concentrations of Ru and Cu in Ru₁Cu SAA are close to the expected concentrations. However, the actual concentration of Ru and Cu in RuCu NPs is different from the expected concentration, mainly because of the metal copper loss involved in the Galvanic replacement process.

Table R1. Ru and Cu precursor concentrations (raw material concentrations) and actual concentrations (ICP-OES test results) of Cu NCs, Ru₁Cu SAA and RuCu NPs.

sample		Cu (wt%)	Ru (wt%)	Cu:Ru (at%)
Cu NCs	precursor concentrations	4.76	-	-
	actual concentrations	5.83	-	-
Ru ₁ Cu SAA	precursor concentrations	4.75	0.36	95.45: 4.55
	actual concentrations	5.31	0.37	95.80: 4.20
RuCu NPs	precursor concentrations	6.82	2.33	82.32: 17.68
	actual concentrations	4.62	2.53	74.36: 25.64

3. The Ru 3d_{5/2} and 3/2 peak ratios in Fig S5 do not seem well aligned with the theoretical ratio. What is the ratio? This should be clarified. XPS data of Ru metal and Ru oxide would be helpful to explain the 0.54 eV shift. The shifts should be explained considering the oxidation state of Ru.

Reply: We appreciate you for this insightful suggestion. We apologize for the misunderstandings caused by our imperfect handling of XPS results.

Firstly, the ratio of Ru 3d_{5/2} and 3d_{3/2} peaks seems to be inconsistent with the theoretical ratio mainly due to the partial overlap of the characteristic binding energy of Ru 3d_{3/2} with the C 1s signal (**Figure R1**). Among them, the two peaks near 284.9 and 286.2 eV in the C 1s & Ru 3d spectra of Cu NCs, Ru₁Cu SAA and RuCu NPs are mainly attributed to the carbon pollution on the catalyst surface (C=C, C-O) (*Nat. Commun.* **13**, 1270 (2022), *Nano Energy* **72**, 104667 (2020)). Because the characteristic binding energy of Ru 3d_{3/2} partially overlaps with the C 1s signal, we mainly studied the Ru 3d_{5/2} XPS signal in this manuscript (*Nat. Commun.* **6**, 6540 (2015), *Energy*

Environment. Sci. **11**, 800-806 (2018)). As shown in **Figure R1**, based on your comments, we have refitted the XPS data for Ru 3d and added a Ru 3d_{3/2} related data analysis (**Supplementary Figure 6a**). The results show that the peak area ratio of Ru 3d_{5/2} and 3d_{3/2} is about 1.58:1, which is close to the theoretical value of 1.5:1.

Figure R1. C 1s & Ru 3d XPS spectra of Cu NCs, Ru₁Cu SAA and RuCu NPs.

Secondly, we agree that the higher-energy shift of the Ru 3d peak can be explained by the oxidation state of Ru. We mentioned in the manuscript that "the binding energy of Ru₁Cu SAA shifts toward high binding energy by ~0.54 eV as compared with that of RuCu NPs, indicating that the Ru species carry more positive charges", which indeed indicated that the Ru in the Ru₁Cu SAA was in an oxidized state. It is mainly caused by the following reasons, the electron transfer between Ru and Cu, the possible oxidation of the catalyst sample when exposed to air, and the interaction between some Ru atoms on the Ru₁Cu SAA and the TiO₂ substrate (*ACS Energy Lett.* **5**, 192-199 (2020), *Adv. Sci.* **10**, 2206096 (2023), *Angew. Chem. Int. Ed.* **61**, e202212542 (2022)).

4. The manuscript must provide a detailed model of the EXAFS spectra. This is vital to

compare the DFT predictions and accuracy of the DFT modeling structures, as well as to give a confident analysis of the EXAFS analysis. The operando EXAFS analysis should also be reanalyzed once the proper model is proposed.

Reply: We are very grateful for the constructive comments. Regarding the fitting of the Ru K-edge R space, we only considered the Ru-Cu shell but not the Ru-O shell for the following reasons: the appearance of the Ru-O shell is due to the inevitable oxidation of Ru exposed to air (*Nat. Catal.* **2**, 304-313 (2019), *Angew. Chem. Int. Ed.* **61**, e202209849 (2022)), and the loading of Ru atoms on the TiO₂ substrate during the material synthesis process. As shown in the **Figure R2a**, it can be found that Cu/Ru in the form of single atoms does exist on the TiO₂ substrate. Combined with Cu K-edge EXAFS results, we speculate that the atoms dispersed on the TiO₂ substrate are mainly Ru atoms.

The effect of Ru on TiO₂ substrate on the performance of electrocatalytic synthesis of formamide: this phenomenon may also exist in RuCu NPs, but the catalytic activity is quite different. We believe that the excellent catalytic activity of Ru₁Cu SAA is mainly caused by the single atomic state of Ru-Cu. To further prove our conjecture, we synthesized Ru/TiO₂ catalyst and tested its performance. The results showed that it exhibited high hydrogen evolution reaction (HER) activity and poor formamide selectivity (**Figure R2b**), which was consistent with our guess. In addition, Ru single atoms on the TiO₂ substrate will also bring great trouble to the establishment of the EXAFS model. Therefore, we chose the model optimized by density functional theory (DFT) as the EXAFS fitting model for data analysis, without taking Ru on the TiO₂

substrate into account.

As shown in **Figure R2c, d** (**Figure 1f, h** in the revised manuscript), we give the specific fitting model and fitting path. We added detailed fitting information as shown in **Table R2** (**Supplementary Table S2** in the revised manuscript).

Figure R2. (a) HAADF-STEM image of Ru₁Cu SAA, Scale bar: 2 nm. (b) Products distribution at different applied potentials in a CO-saturated 1 M KOH + 1 M KNO₂ solution on Ru/TiO₂. (c) Ru K-edge FT-EXAFS spectra of Ru₁Cu SAA, RuO₂, and Ru foil, and corresponding Ru₁Cu SAA EXAFS fitting curves, inset showing the schematic model. (d) Cu K-edge FT-EXAFS spectra of Ru₁Cu SAA, Cu₂O, CuO, and Cu foil, and corresponding Ru₁Cu SAA EXAFS fitting curves, inset showing the schematic model.

Table R2. EXAFS fitting parameters at the Ru K-edge and Cu K -edge for Ru₁Cu SAA.

	Path	CN	R (Å)	$\sigma^2(10^{-3} \text{ Å}^2)$	$\Delta E0$ (eV)	R-factor
Ru K-edge	Ru-Cu	4.8 (± 0.3)	2.67 (± 0.03)	6.6 (± 0.3)	7.0	0.01
Cu K - edge	Cu-Cu	8.0 (± 0.3)	2.53 (± 0.01)	9.1 (± 1.0)	4.5	0.006

5. The authors stated that a primitive model and single path fitting are provided in Fig S7 and Table S2. How the EXAFS model was chosen is unclear, and Table S2 is confusing. The first and second shell scattering paths should be provided.

Reply: We appreciate you for the constructive comment and suggestion. As shown in **Figure R2** and **Table R2**, we give the specific fitting model, fitting path, and detailed fitting information (Please see **comment 4**).

6. The comparison of the DFT model with the EXAFS analysis must be provided in the manuscript. The proposed DFT structures should be used to model the EXAFS spectra.

Reply: We greatly appreciate your kind reminders and suggestions. The model selected for EXAFS fitting is the model obtained by DFT optimization.

7. The error bars for the EXAFS fitting must be provided.

Reply: We thank the reviewer for the helpful suggestions. Based on your suggestion, we have added corresponding error bars to the EXAFS fitting parameters table (Please see **Table R2**).

8. When properly fitted, the reviewer recommends showing the results in the manuscript, not the supporting information, which is essential data that will combine the EXAFS and DFT analysis.

Reply: We appreciate you for this insightful comment. Following the suggestions, we present the EXAFS fitting results of Ru K-edge and Cu K-edge of Ru₁Cu SAA (**Figure R2c, d**), and give the corresponding fitting parameters in **Table R2** (Please see **comment 4**).

9. The EXAFS analysis of RuCu nanoparticles should be provided.

Reply: We appreciate the reviewer for this reminding. We agree that the use of RuCu NPs as a reference helps us to further confirm the coordination environment of Ru₁Cu SAA. However, affected by the Corona Virus Disease 2019 (COVID-19), many express logistics centers are closed and the researchers cannot return to the National Synchrotron Radiation Research Center (NSRRC) of Taiwan this moment. Inspired by this comment, we are trying to conduct this analysis, but at this moment we are unable to share the XAS tests for the reviewer. Thank you for the understanding.

10. The Ru-O bonds observed in the EXAFS spectra are not considered in the DFT model. The effect of the Ru-O oxygen bond, evidenced in the EXAFS, questions the accuracy of the DFT structure.

Reply: We appreciate you for this insightful and constructive comments. The appearance of slight Ru-O bonds is mainly caused by the inevitable oxidation of the material in the air and the loading of some Ru atoms on the TiO₂ substrate during the material synthesis process. The experimental results show that the excellent catalytic activity of Ru₁Cu SAA is mainly caused by the single atomic state of Ru-Cu (Please see **comment 4**). Therefore, during the modeling process, we did not consider the Ru-O bonds when building the model.

11. The STEM EDX image shows that the Ru clusters are forming. Most likely, the structures combine RuCu SAAs and Ru metal clusters. This can be normal. What is the evidence that Ru metals are not forming in these structures? This must be explained thoroughly in the manuscript.

Reply: We appreciate you for this insightful comment. The STEM-coupled energy dispersive spectroscopy (EDS) composition line profile shows the elemental information of a single Ru₁Cu SAA particle, and the signal-to-noise ratio of the obtained information is poor due to the small size of Ru₁Cu SAA and substrate interference. Regarding whether Ru metal was formed, we inferred mainly from the following characterization results: 1) In the XRD patterns of Ru₁Cu SAA, we did not find the presence of diffraction peaks of Ru or RuO₂ phases, which implies a high dispersion of Ru species (**Supplementary Figure 2a** in the revised manuscript). 2) Neither high-angle annular dark-field scanning transmission electron microscopy (HAADF-STEM) images (**Figure 1c** in the revised manuscript) nor STEM-coupled energy-dispersive spectroscopy (EDS) elemental mapping (**Figure 1e** in the revised manuscript) revealed aggregation of Ru atoms. 3) Moreover, the Ru-Ru peak (2.39 Å) does not appear in the corresponding Fourier transform extended X-ray absorption fine structure (FT-EXAFS) spectrum of Ru₁Cu SAA (**Figure 1f** in the revised manuscript). Meanwhile, the wavelet transform (WT) of Ru EXAFS shows different Ru-Cu scattering ($\sim 8.1 \text{ \AA}^{-1}$) from that of Ru foil (**Figure 1g** in the revised manuscript). Combining the above results, we deduce that there is no Ru metal formed in these structures.

12. The authors state, “It is noteworthy that these isolated Ru atoms are surrounded by Cu atoms in different regions of the TiO₂ nanowires, rather than Ru nanoclusters.” It is visible that Ru nanoclusters are also forming. This should be clarified in the manuscript.

Reply: We thank you for your constructive comments. We apologize for the misunderstanding caused by our inappropriate description. We have made relevant changes in the revised manuscript (page 4, line 18-20). What we want to express is that these isolated Ru atoms are dispersed on Cu nanoparticles without agglomeration to form Ru nanoclusters. Moreover, combining the XRD, STEM and XAS characterization results, we speculate that Ru nanoclusters are not present in Ru₁Cu SAA.

13. The reaction coordinate system is determined through the binding energies in DFT simulations. The rate-limiting steps are also determined by the energy difference. This is not necessarily an objective approach. The transition states were not calculated. Activation energies are unknown. Therefore, the proposed reaction mechanism must be justified.

Reply: We appreciate you for the insightful and helpful comment. For this comment, we performed a kinetic transition state (TS) study to further assess the plausibility of the reaction pathway (page 12, line 21-22 page 13, line 1-3, and **Supplementary Figure 30** in the revised manuscript). The kinetic barriers of the transition states of *CO*NOOH → *CO*NOHOH (RDS) and *CO*N → *CO*NH during the formamide electrosynthesis of Ru₁Cu SAA and Cu NC were calculated (**Figure R3**). The kinetic barriers for the formation of *CO*NOHOH and *CO*NH on Ru₁Cu SAA are 0.67 and

0.91 eV respectively. However, the energy barriers are 1.05 and 1.03 eV on Cu NC, which is higher than Ru₁Cu SAA. The calculations based on the relative kinetic barriers of the transition states prove that the formation of *CO*NOHOH and *CO*NH is more favorable in Ru₁Cu SAA from the perspective of energy. It is further demonstrated that the introduction of highly dispersed Ru atoms can effectively enhance the activation hydrogenation step of NO₂⁻.

Figure R3. Kinetic energy barrier of *CO*NOHOH and *CO*NH formation on Ru₁Cu SAA and Cu NCs, respectively.

Reviewer #2 (Remarks to the Author):

Lan et al. report the synthesis of a Ru₁Cu single-atom alloy electrocatalyst and its application for converting CO and nitrite into formamide. High Faradaic efficiencies up to 46% could be achieved for formamide production. Both operando XAS and in situ Raman measurements reveal the electrocatalytic processes of the C-N bond formation on catalyst surface. DFT calculations suggest that CO adsorption at Ru adjacent Cu sites and nitrite reduction at Ru sites can generate the C-N bond. The electroreduction of CO and nitrite to formamide is interesting, and it has not been reported in literature. However, there are quite a few technical problems, which should

be addressed before considering publication in the journal.

Reply: We appreciate the reviewer for recognizing the originality and importance of our work. We also thank the reviewer for the professional comments and suggestions, which are very valuable for improving the scientific impact of this work. By following the reviewer's comments and suggestions, we carefully revised the manuscript and clarified the reviewer's comments. The details will be described below.

1. The equations for calculating the Faradaic efficiency are problematic. The authors should also provide the detailed data to calculate FE of formamide.

Reply: We thank the reviewer for careful reading our paper. We found and corrected related errors in the revised manuscript (page18, lines 6-9). Meanwhile, the Faradic efficiency for formamide was calculated as follows:

$$\text{Faradaic efficiency (\%)} = (nFCV)/Q * 100$$

Where n is the number of charge transfers (n = 6); V is electrolyte volume; C is the concentration of formamide; F is the Faraday constant (F = 96,485.3 C mol⁻¹); Q is the electric quantity.

2. TiO₂ nanowires were used as the support for the catalyst, why? How about the conductivity of TiO₂ nanowires?

Reply: We thank the reviewer for these nice comments and questions. Metal oxides are considered ideal supports for designing catalysts (*Adv. Mater.* **30**, 1705369 (2018)). Among them, TiO₂ is one of the most widely studied metal oxide materials. It is widely used to support single atoms, support nanoparticles, support alloy nanoparticles, high-entropy alloy nanoparticles, etc.; and has a wide range of industrial applications (*Nat.*

Commun. **11**, 48 (2020), *Nat. Commun.* **10**, 5790 (2019), *Nat. Chem.* **12**, 717-724 (2020), *Nat. Commun.* **12**, 3884 (2021)). TiO₂ as a substrate has the following advantages (*Energy Environ. Sci.* **7**, 2535-2558 (2014), *J. Mater. Chem. A* **4**, 14-31 (2016), *Chem. Rev.* **107**, 2891-2959 (2007), *Adv. Mater.* **30**, 1705369 (2018)): Firstly, TiO₂ has intriguing catalytic properties and excellent stability in both acidic and alkaline solutions. Secondly, there is a strong interaction between TiO₂ and metal nanoparticles, which can avoid agglomeration of metal particles, disperse metal atoms in clusters, and change the electronic properties of nano-metal catalysts. Thirdly, the structure of TiO₂ is controllable, and different structures can be synthesized according to requirements, such as nanotubes, nanorods, nanofibers and nanosheets, etc. Fourthly, TiO₂, as a reducible metal oxide, can tailor material properties, such as optical and electrical properties.

Regarding the conductivity of TiO₂, although TiO₂ as a carrier or catalyst exhibits certain catalytic properties, compared with pure metals, the electronic conductivity of TiO₂ is relatively low. However, studies have shown that the electronic conductivity of TiO₂ can be effectively improved by creating oxygen vacancies and titanium gaps, as well as doping other metals (*Energy Environ. Sci.* **7**, 2535-2558 (2014)). For this opinion, we tested the oxygen vacancies of Ru₁Cu SAA by electron paramagnetic resonance (EPR) spectroscopic characterization (**Figure R4**), which showed an abundance of oxygen vacancies. In addition, the existence of defects can promote the electrocatalytic process. We are also interested in the effect of defects on the synthesis of formamide, but this is not the focus of this paper. We hope to study it in detail in the

follow-up work.

Figure R4. EPR spectra of pristine Ru₁Cu SAA.

3. The electron microscopy characterizations need improvement. The SEM image in Fig 1b is not clear. Figs 1c and 1d could not show clearly single Ru atom. Ru mapping in Fig 1e does not match well with the Cu mapping.

Reply: We appreciate you for the constructive comment and suggestion. We apologize for the imperfect electron microscopy characterization. In this regard, we made corresponding improvements to the electron microscopy characterization. We re-updated a clearer SEM picture (**Figure R5a**). Due to the small particle size of Ru₁Cu SAA, the atomic numbers of Ru and Cu are relatively close, and there is little difference in atomic contrast, so the individual Ru atoms on Ru₁Cu SAA particles are not obvious. For this purpose, we sharpen HAADF-STEM accordingly (**Figure R5b**). In addition, we re-updated a clearer enlarged image of a single Ru₁Cu SAA particle (**Figure R5c**), which is more obvious than the original image of a single Ru atom. Due to the small size of the element mapping, the details of the image may be missing. As shown in **Figure R5d**, after zooming in, it can be found that the Ru mapping basically matches

the Cu mapping.

Figure R5. (a) SEM image of the Ru₁Cu SAA. (b) HAADF-STEM image of Ru₁Cu SAA (c) Magnified image of a single Ru₁Cu SAA particle. (d) HAADF-STEM image and the corresponding elemental mapping. Scale bars: a 200 nm, b 2 nm, c 1 nm, d 5 nm.

4. Some control reactions, for example the coreduction of: CO+nitrate, CO+NO, CO+hydroxylamine, CO+amine, should be conducted to better understand the C-N coupling process.

Reply: We appreciate you for this insightful and constructive suggestion. In the experimental stage, we tried different nitrogen sources (NO₃⁻, NO₂⁻, NH₃·H₂O) for C-N coupling experiments. As shown in **Figure R6a**, formamide could also be obtained when performing C-N coupling experiments using NO₃⁻ as N source. However, a large amount of by-products (NO₂⁻) were produced during the reaction (**Figure R6b**), and the poor selectivity of formamide is observed. We found that both NO₃⁻ and NO₂⁻ could be used for coupling experiments with CO to obtain formamide products, so it is

speculated that using NO or N₂ as a nitrogen source for coupling experiments may also obtain formamide.

Figure R6. (a) ¹H NMR spectra of the electrolyte obtained after co-reduction with NO₃⁻ and NH₃ H₂O with CO, respectively. (b) Products distribution at different applied potentials in a CO-saturated 1 M KOH + 1 M KNO₃ solution on Ru₁Cu SAA.

However, when coupling experiments were performed using NH₃·H₂O as N source with CO, no observable formamide appeared from the ¹H NMR spectrum of the reacted electrolyte (**Figure R6a**). Surprisingly, a new NMR peak appeared, which may correspond to a new organic nitride. We think this is a very interesting experiment, and it is possible to draw new research conclusions and further promote the progress in the field of C-N coupling synthesis of organic nitrides, which will appear in our follow-up research work.

Combined with the above experimental results, we speculate that *NH₂ is an important intermediate for the synthesis of formamide. Therefore, we did not consider further C-N coupling experiments using hydroxylamines and amines as nitrogen sources. However, we are very interested in further exploring organic nitrogen species using hydroxylamine or amines as nitrogen sources, which may arise in our follow-up

work.

5. The current density in the stability test (Supplementary Fig. 19, -10 mA/cm^2) is much smaller than the LSV measurement (Supplementary Fig. 14, -20 mA/cm^2 at -0.5 V), why?

Reply: We appreciate you for this insightful comment. The current density of LSV differs from that of chronoamperometry, which may be attributed to the following three factors. First, LSV testing is a transient process, while chronoamperometric testing is a steady-state process. Second, double-layer charging, and competing HER hydrogen production on the catalyst surface hinder the contact of the catalyst with the solution (*Nat. Commun.* **9**, 3485 (2018)). Third, it is caused by experimental system errors.

Reviewer #3 (Remarks to the Author):

Title: Efficient Electrosynthesis of Formamide from Carbon Monoxide and Nitrite on a Ru-dispersed Cu nanocluster catalyst

This manuscript reported the electrochemical process for the synthesis of high-valued formamide from carbon monoxide and nitrite with a Ru_1Cu single-atom alloy under ambient conditions. Operando X-ray absorption spectroscopy, in situ Raman spectroscopy and density functional theory calculations results were used to detect intermediates. However, there are still some problems in the material characterization and properties investigation. Especially, the reaction mechanism is not clear.

Reply: We appreciate the reviewer for his/her valuable comments/suggestions. Following the comments and suggestions, we have conducted additional experiments and analyses and carefully revised the manuscript. The details will be described below.

1. Figure 1e is too fuzzy to distinguish Cu from Ru. The Electron energy loss

spectroscopy (EELS) is better to distinguish Cu and Ru than HAADF-STEM image and the corresponding elemental mapping.

Reply: We appreciate you for the constructive comment and suggestion. For this comment, we have re-uploaded a clearer zoom-in image of a single Ru₁Cu SAA particle (**Figure R7**), in which individual Ru atoms are more clearly shown compared to the original image in the unrevised manuscript. We also believe that electron energy loss spectroscopy (EELS) is indeed a very effective means of distinguishing different atoms. However, during the previous characterization process, we found that the EELS test results were not satisfactory. This may be caused by the following reasons: 1) due to the small difference in atomic contrast between Cu and Ru, it is difficult to effectively distinguish the two types of atoms; 2) because Ru₁Cu SAA is a nano-cluster, the number of atomic layers is large and the size is thick, multiple scattering is serious, and the effect is not well (*Adv. Mater.* **30**, 1706755 (2018), *J. Appl. Phys.* **99**, 084902 (2006), *Chem* **8**, 3008-3017, (2022)).

Figure R7. Magnified image of a single Ru₁Cu SAA particle. Scale bar: 1 nm.

2. The authors adopted Galvanic replacement strategy to prepare Ru₁Cu single-atom alloy. What is the driving force for the formation of Ru₁Cu single-atom alloy? The

Ru₁Cu single-atom structure is uniform? If not, how the authors chose the model for simulation?

Reply: We appreciate you for this insightful comment. Galvanic replacement is the spontaneous replacement of surface layers of a metal (M), by a more noble metal (M_{noble}), when the former is treated with a solution containing the latter in ionic form, according to the general replacement reaction: $nM + mM_{\text{noble}}^{n+} \rightarrow nM^{m+} + mM_{\text{noble}}$. The reaction is driven by the difference in the equilibrium potential of the two metal/metal ion redox couples (*Adv. Mater.* **25**, 6313-6332 (2013), *Catalysts* **7**, 80 (2017), *Nano Today* **34**, 100917 (2020)).

However, it is really not easy to characterize whether the structure of each Ru₁Cu SAA particle is uniform from a microscopic point of view. Regarding whether the Ru₁Cu SAA structure is uniform, we mainly speculate from the macroscopic characterization results. Based on the macroscopic characterization results (XRD, XPS, XAS, ICP-OES), it can be inferred that the Ru atoms on the Cu nanoclusters present an atomic dispersion state.

3. As the authors say that the electrochemical synthesis of organic amides from C-N coupling reaction by introducing an NH₃ source during CO₂/CO reduction reaction, the formamide detected in this work may be formed by the reaction of NH₃ and CO. The labeled nitrite and unlabeled ammonia can be used as N-source to explore the origin of N under reductive conditions.

Reply: We appreciate you for this insightful and constructive comment. First, we performed coupled experiments using labeled nitrite (¹⁵NO₂⁻) and labeled CO (¹³CO)

to explore the source of N/C under reducing conditions (page 8, line 2-7, and **Figure 2c, d** in the revised manuscript). As shown in **Figure R8a**, the coupling reaction of $^{15}\text{NO}_2^-$ and ^{13}CO resulted in typical $\text{H}^{13}\text{CO}^{15}\text{NH}_2$ peaks at 7.7 and 8.2 ppm in ^1H NMR spectrum of the electrolyte. In contrast, the ^1H NMR spectrum of the electrolyte after the coupling reaction of $^{14}\text{NO}_2^-$ and ^{12}CO only shows a typical peak of $\text{H}^{12}\text{CO}^{14}\text{NH}_2$ at 8.3 ppm (**Figure 2c** in the revised manuscript). In order to further confirm the production of $\text{H}^{13}\text{CO}^{15}\text{NH}_2$, we also performed GC-MS testing. The results show that the M/Z of the obtained electrolyte is consistent with the standard $\text{H}^{13}\text{CO}^{15}\text{NH}_2$ (**Figure R8b**). The above results confirmed that the generated HCONH_2 originated from the electrocatalytic coupling reaction of CO and NO_2^- on Ru_1Cu SAA.

Figure R8. (a) ^1H NMR spectra of Ru_1Cu SAA with $^{15}\text{NO}_2^-$ and ^{13}CO as nitrogen sources along with standard references ($\text{H}^{13}\text{CO}^{15}\text{NH}_2$). (b) GC-MS results of the electrolyte obtained after $^{15}\text{NO}_2^-$ and ^{13}CO co-reduction. (c) ^1H NMR spectra of the electrolyte obtained after the $\text{NH}_3 \cdot \text{H}_2\text{O}$ and CO co-reduction.

Moreover, we also considered using $\text{NH}_3 \cdot \text{H}_2\text{O}$ as N source to react with CO. However, when coupling experiments were performed using $\text{NH}_3 \cdot \text{H}_2\text{O}$ as N source with CO, no observable formamide appeared from the ^1H NMR spectrum of the reacted

electrolyte (**Figure R8c**). Surprisingly, a new NMR peak appeared, which may correspond to a new organic nitride. We think this is a very interesting experiment, and it is possible to draw new research conclusions and further promote the progress in the field of C-N coupling synthesis of organic nitrides, which will appear in our follow-up research work.

4. As shown in Figure 2a, the main C-related product is CH₃COOH (FE>80%), why is acetamide not detected in this work?

Reply: We sincerely thank you for this comment. As shown in **Figure R9**, the products of the CO reduction reaction of Ru₁Cu SAA were mainly hydrogen gas and a small amount of acetic acid. Due to the poor selectivity of Ru₁Cu SAA for acetic acid, we did not detect observable acetamide in the reactants.

Figure R9. Faradaic efficiencies of major reduction products on Ru₁Cu SAA for CORR.

5. In Figure 2c, why is the H₂O peak of NO₂⁻+CO in ¹H NMR a downward inverted peak?

Reply: We thank the reviewer for this nice comment. We apologize for the misunderstanding caused by our improper data handling. We truncated 3-6.5 ppm (truncated the water peak (~4.8 ppm)) during the data processing. As shown in **Figure**

R10a, the water peak is normal in the NMR full spectrum. To avoid unnecessary misunderstandings, we have corrected the **Figure R10b** (**Figure 2c** in revised manuscript).

Figure R10. (a) Complete ¹H NMR spectrum of a standard formamide sample. (b) ¹H NMR spectra of standard formamide sample and the electrolyte obtained after CORR, NO₂⁻RR and co-reduction of CO and NO₂⁻.

6. In Figure 3g, the baseline of the -0.5 V spectrum is very uneven. The formamide has been formed in all potentials, why the intermediates can be only detected at -0.5 V? Isotope labeling experiments must be provided to better confirm the intermediates.

Reply: We appreciate you for this important comment and reminding. The in-situ Raman test used in this paper is conventional Raman spectroscopy, and its application in water environments will face the following problems: low detection sensitivity, poor anti-interference ability, interference by air bubbles, etc (*Nat. Commun.* **13**, 2656 (2022), *Nat. Commun.* **8**, 15447 (2017)). Therefore, the signal-to-noise ratio of the obtained spectral information is poor. Furthermore, we apologize for the inconvenience caused by the inappropriate data processing. Therefore, we reprocessed the in situ Raman data. As shown in the **Figure R11**, it can be found that significant intermediate

stretching vibration peaks can be observed under the condition of -0.3 - 0.7 V vs. RHE.

Figure R11. In-situ Raman spectra of Ru₁Cu SAA at various potentials in a CO-saturated 1 M KOH + 1 M KNO₂ solution.

We performed the isotope labeling experiments (Please see **Comment 3**), confirming that the generated HCONH₂ comes from the electrocatalytic coupling reaction of CO and NO₂⁻ on Ru₁Cu SAA, which also proves that both C/N in the intermediates are derived from the reactants. We believe that the characterization of the isotope-labeled nitrite (¹⁵NO₂⁻) and labeled CO (¹³CO) coupling experimental product will also confirm the origin of the C/N in the intermediate.

7. Stability is an important parameter to evaluate the materials. The authors should carry out the long-term test under high current density. In addition, the characterization of the catalytic materials after test should be provided.

Reply: We appreciate you for this insightful suggestion. Following this comment, we conducted high current density stability tests and performed XRD and SEM tests on the materials after the stability tests (Page 9, line 4-6, and **Supplementary Figure 21** in

the revised manuscript). The high current density stability test results show that the voltage required to reach 250 mA cm^{-2} on the Ru_1Cu SAA catalyst remains stable for 50 h (**Figure R12a**). At the same time, after the high current density stability test, the phase (**Figure R12b**) and microstructure (**Figure R12c**) of Ru_1Cu SAA did not change significantly. The above results demonstrate the excellent high current density stability of Ru_1Cu SAA for formamide electrosynthesis.

Figure R12. (a) Stability test results using the MEA electrolyzer at a total current density of 250 mA cm^{-2} for 50 hours. (b, c) The XRD (b) and SEM (c) image after high current density stability test, Scale bar: 200 nm.

8. The author should carefully check the horizontal coordinate of Figure 4a and should

not make this kind of mistake.

Reply: We thank the reviewers for their careful reading of our paper. We have corrected related issues in **Figure 4a** of the revised manuscript. As shown in the **Figure R13**, we adjusted the horizontal coordinate labeling and marked the specific reaction steps in the figure.

Figure R13. Diagram of free energy changes for NO_2^- reduction on Ru_1Cu SAA surface with the assistance of $^*\text{CO}$. Insert shows free-energy diagram for NO_2^- RR on Ru_1Cu SAA and Cu NCs.

9. The resolution of all the pictures in the article is too low, which affects the reader's reading.

Reply: We thank the reviewer for the helpful suggestions. We updated all images and increased the resolution of the images.

Reviewer #4 (Remarks to the Author):

In this paper, the authors prepare atomically dispersed Ru atoms on Cu nanoclusters single atom alloy loaded on TiO_2 nanowires. By using this catalyst, they conduct

electroreduction of CO and NO₂⁻ to product formamide with high Faradaic efficiency. This work offers a new insight into the synthesis of organic nitrogen compounds under ambient conditions. However, several issues should be addressed before I can recommend publication. The following suggestions are provided for the authors:

Reply: We appreciate the reviewer for recognizing the originality and importance of our work. We also thank the reviewer for the deep and professional comments and suggestions, which are very valuable for improving the scientific impact of this work. By following the reviewer's comments and suggestions, we explained and described in detail. The details will be described below.

1. TiO₂, as a widely used semiconductor in photocatalysis. Why the authors choose TiO₂ as the catalyst support?

Reply: We thank the reviewer for these nice comments and questions. As you said, TiO₂ is indeed a semiconductor material widely used in photocatalysis. Likewise, it is also one of the most extensively studied electrocatalyst supports. It is widely used to support single atoms, support nanoparticles, support alloy nanoparticles, high-entropy alloy nanoparticles, etc., and has a wide range of industrial applications (*Nat. Commun.* **11**, 48 (2020), *Nat. Commun.* **10**, 5790 (2019), *Nat. Chem.* **12**, 717-724 (2020), *Nat. Commun.* **12**, 3884 (2021)). TiO₂ as a substrate has the following advantages (*Energy Environ. Sci.* **7**, 2535-2558 (2014), *J. Mater. Chem. A* **4**, 14-31(2016), *Chem. Rev.* **107**, 2891-2959 (2007), *Adv.Mater.* **30**, 1705369(2018)): Firstly, TiO₂ has excellent stability in both acidic and alkaline solutions. Secondly, there is a strong interaction between TiO₂ and metal nanoparticles, which can avoid agglomeration of metal particles, disperse metal atoms in clusters, and change the electronic properties of nano-metal

catalysts. Thirdly, the structure of TiO₂ is controllable, and different structures can be synthesized according to requirements, such as nanotubes, nanorods, nanofibers and nanosheets, etc. Fourthly, TiO₂, as a reducible metal oxide, can tailor material properties.

2. In lines 204-208, “a remarkable positive shift under OCV condition compared with that under EX situ condition” and “the emerging Cu-O/N/C shell scattering peaks in the corresponding FT-EXAFS spectra” are ascribed to the adsorbed reactants on the Cu atoms. Authors need to provide more convincing evidences.

Reply: We appreciate you for this insightful and constructive comment. Previous studies have proven that the absorption edge of the K edge in the XANES spectrum moves to a higher energy side under OCV conditions, which is mainly caused by the delocalization of electrons caused by the adsorption of reactants (*Nat. Catal.* **2**, 134-141 (2019), *Nat. Commun.* **12**, 1687 (2021)). In addition, this is also confirmed by the appearance of new shell scattering in the FT-EXAFS spectra. The reactants in the solution are H₂O, NO₂⁻, CO. We believe that analyzing the specific reactants adsorbed on the catalyst is crucial to clarify the mechanism of C-N coupling catalysis. Unfortunately, the adsorption state is not yet clear, which makes modeling difficult and difficult for path fitting. However, we investigated the CO adsorption capacity of Ru₁Cu SAA by CO-TPD test. The results show that Ru₁Cu SAA presented the strongest CO chemisorption (**Figure R14** and **Table R3**), which is higher than the precursor TiO₂ nanowires, Cu NCs and RuCu NPs, indicating that Ru₁Cu SAA has a good adsorption capacity for CO.

Figure R14. CO-TPD spectra of TiO₂ nanowires, Cu NCs, Ru₁Cu SAA, and RuCu NPs.

Table R3. The CO-TPD test results.

Sample	quality	total peak area	CO adsorption capacity (* 10 ⁻⁵ mol)	CO specific mass adsorption (mmol g ⁻¹)
Cu NCs	0.05	2980	2.93	0.58
Ru ₁ Cu SAA	0.05	4905	4.82	0.95
RuCu NPs	0.05	2050	2.01	0.40
TiO ₂ nanowires	0.05	1530	0.95	0.19

3. In line 224, “CO stretching at 1890 and 2903 cm⁻¹”, this value in the text was mismatch with the results in Fig. 3g.

Reply: We thank the reviewers for carefully reading our paper and pointing out our errors. We have corrected related errors (CO stretching at the 1890 and 2190 cm⁻¹) in the revised manuscript (page 11, line 14).

4. In Supplementary Fig. 20, XRD patten of Ru₁Cu SAA before stability test should be provided for better comparison.

Reply: We thank the reviewer for the helpful suggestions. Based on your suggestion,

we have added the Ru₁Cu SAA XRD pattern before the stability test (**Figure R15**). However, we deleted **Supplementary Fig. 20** in the revised manuscript and replaced it with the XRD pattern after high current stability testing (**Figure R12b**).

Figure R15. The XRD pattern of Ru₁Cu SAA before and after stability test.

5. As important control samples-RuCu NPs, the authors need to supply more basic characterizations such as SEM, HAADF-STEM.

Reply: We appreciate your very constructive suggestions. Following this comment, we further performed the characterizations of RuCu NPs (Page 5, line 4-7, and **Supplementary Figure 5** in the revised manuscript). As shown in **Figure R16a**, the HAADF-STEM image shows that RuCu NPs are uniformly dispersed on the TiO₂ nanowire support with an average size of about 1.1 nm. The energy dispersive spectroscopic (EDS) elemental mapping (**Figure R16b**) analysis further revealed the simultaneous presence and uniform dispersion of Cu and Ru, indicating a uniform composition of the particles. Furthermore, **Figure R16b** shows that the Ru content of RuCu NPs is significantly higher than that of Ru₁Cu SAA, matching the experimental design.

Figure R16. (a) HAADF-STEM image of RuCu NPs. (d) corresponding enlarged images. Scale bars: (a) 5 nm, (b) 1 nm

6. Does the nitrite concentration affected on the C-N coupling process?

Reply: We appreciate you for the insightful comment. For this comment, we investigated the effect of different concentrations of nitrite on the course of the C-N coupling reaction. During the experiment, we used different concentrations of nitrite (0.01M, 0.1M, and 1M) for the coupling reaction with CO. As shown in **Figure R17a**, under the condition of low nitrite concentration, the electrosynthesis of formamide can also be realized. The main side reactions in the formamide synthesis process are HER and ammonia synthesis. As shown in **Figure R17b**, as the concentration of nitrite decreases, the competitive HER reaction is gradually enhanced, while the selectivity of synthetic ammonia is gradually weakened. Considering that the selectivity of the side reaction ammonia synthesis gradually increases with increasing nitrite concentration, this will inevitably lead to a decrease in formamide selectivity. Therefore, we did not consider to further increase the nitrite concentration.

Figure R17. (a) ^1H NMR spectra of electrolytes obtained after co-reduction of CO with different concentrations of KNO_2 . (b) The Faradaic efficiencies of H_2 and NH_3 in the co-reduction products of CO with different concentrations of KNO_2 at -0.5 V vs. RHE.

7. The usage of “Operando” should be strict.

Reply: We sincerely thank you for this comment. We have carefully checked the manuscript and revised relevant issues.

Minor suggestion: Some bottom of the words in some figures are covered, such as Fig. 2, Supplementary Fig. 6 and Supplementary Fig. 11b.

Reply: We thank the reviewers for their careful reading of our paper. We have carefully checked all images and corrected relevant issues.

REVIEWERS' COMMENTS

Reviewer #1 (Remarks to the Author):

The authors performed necessary changes in the manuscript and addressed reviewer comments. The reviewer recommends no further revisions.

Reviewer #3 (Remarks to the Author):

I appreciate the substantial efforts made by the authors. After carefully examining the revised manuscript, I found that the authors have fully addressed all of the points that I have brought up, as well as the concerns raised by the other reviewers. The quality of the revised manuscript has been greatly improved, but there are still a few details to note.

The peak position of HCONH₂ is located at 8.3 ppm in this paper, which is similar to the peak position of HCOOH on CO₂ reduction field. Why? The ¹H NMR test details should be provided.

Reviewer #4 (Remarks to the Author):

The article submitted by Tan et al. is a revised version which I had reviewed previously. The article has been improved. I have no further question.

Responses to the Referees' Comments

We would like to thank all the referees for the careful review and the valuable comments.

We have carefully considered the referees' comments and revised the manuscript accordingly. Below we list the changes we have made in light of the referees' comments.

Reviewer #1 (Remarks to the Author):

The authors performed necessary changes in the manuscript and addressed reviewer comments. The reviewer recommends no further revisions.

Reply: We are very grateful to your encouraging and positive comments and really appreciate your agreement of acceptance with this revised manuscript.

Reviewer #3 (Remarks to the Author):

I appreciate the substantial efforts made by the authors. After carefully examining the revised manuscript, I found that the authors have fully addressed all of the points that I have brought up, as well as the concerns raised by the other reviewers. The quality of the revised manuscript has been greatly improved, but there are still a few details to note.

Reply: We appreciate your positive comments and acknowledgement of the work we have done. Based on your suggestions, we have conducted a more appropriate experimental analysis and carefully revised the manuscript.

1. The peak position of HCONH_2 is located at 8.3 ppm in this paper, which is similar

to the peak position of HCOOH on CO₂ reduction field. Why? The ¹H NMR test details should be provided.

Reply: We appreciate you for this insightful and constructive comment. We also considered this issue at the beginning of our experiment, so we conducted a series of validation experiments (such as ¹H NMR, GC-MS, and blank control experiments) to confirm the synthesis of formamide. The ¹H NMR peak of formamide is very close to the ¹H NMR peak of formic acid, which may be caused by the high pH of the solution. As shown in **Figure R1**, we can observe that the position of the NMR peak of formamide changes significantly with the increase of pH. It is precisely because the ¹H NMR peak of formamide is very close to that of formic acid. Therefore, we did not use the NMR results as the main basis for formamide production, but further conducted GC-MS testing. Based on your suggestion, we have added details of relevant NMR testing to the revised manuscript (page 17, line 5-8).

Figure R1. ¹H NMR spectra of formamide in solutions with different pH.

Reviewer #4 (Remarks to the Author):

The article submitted by Tan et al. is a revised version which I had reviewed previously.

The article has been improved. I have no further question.

Reply: We appreciate your recommendation of acceptance and helpful comments in the reviewing process and are pleased to have our manuscript be reviewed by you.